# Hepatocyte-specific loss of GPS2 in mice reduces non-alcoholic steatohepatitis via activation of PPARα

Ning Liang[1], Anastasius Damdimopoulos[1], Saioa Goñi[1], Zhiqiang Huang[1], Lise-Lotte Vedin [2], Tomas Jakobsson[2], Marco Giudici[1], Osman Ahmed [2], Matteo Pedrelli[2], Serena Barilla[1], Fawaz Alzaid [3], Arturo Mendoza[4], Tarja Schröder[2], Raoul Kuiper[2], Paolo Parini[2,5,6], Anthony Hollenberg[4], Philippe Lefebvre [7], Sven Francque [8,9], Luc Van Gaal[9,10], Bart Staels[7], Nicolas Venteclef[3], Eckardt Treuter[1] & Rongrong Fan[1]

Obesity triggers the development of non-alcoholic fatty liver disease (NAFLD), which involves alterations of regulatory transcription networks and epigenomes in hepatocytes. Here we demonstrate that G protein pathway suppressor 2 (GPS2), a subunit of the nuclear receptor corepressor (NCOR) and histone deacetylase 3 (HDAC3) complex, has a central role in these alterations and accelerates the progression of NAFLD towards non-alcoholic steatohepatitis (NASH). Hepatocyte-specific *Gps2* knockout in mice alleviates the development of diet-induced steatosis and fibrosis and causes activation of lipid catabolic genes. Integrative cistrome, epigenome and transcriptome analysis identifies the lipid-sensing peroxisome proliferator-activated receptor α (PPARα, NR1C1) as a direct GPS2 target. Liver gene expression data from human patients reveal that *Gps2* expression positively correlates with a NASH/fibrosis gene signature. Collectively, our data suggest that the GPS2-PPARα partnership in hepatocytes coordinates the progression of NAFLD in mice and in humans and thus might be of therapeutic interest.

[1] Department of Biosciences and Nutrition, Karolinska Institutet, Huddinge 14157, Sweden. [2] Department of Laboratory Medicine, Karolinska Institutet, Huddinge 14157, Sweden. [3] INSERM, Cordeliers Research Centre, Sorbonne Paris Cité, Université Paris Descartes, Université Paris Diderot, Paris 75013, France. [4] Division of Endocrinology, Diabetes and Metabolism, Joan & Sanford I. Weill Department of Medicine, Weill Cornell Medicine, New York 10021, USA. [5] Department of Medicine Huddinge, Karolinska Institutet, Huddinge 14157, Sweden. [6] Inflammation and Infection Theme, Karolinska University Hospital, Huddinge 14157, Sweden. [7] University Lille, INSERM, CHU Lille, Institut Pasteur de Lille, U1011-EGID, Lille F-59000, France. [8] Department of Gastroenterology and Hepatology, University of Antwerp, Antwerp 2610, Belgium. [9] Laboratory of Experimental Medicine and Pediatrics, University of Antwerp, Antwerp 2610, Belgium. [10] Department of Endocrinology, Diabetology and Metabolism, University of Antwerp, Antwerp 2610, Belgium. Correspondence and requests for materials should be addressed to E.T. (email: eckardt.treuter@ki.se) or to R.F. (email: rongrong.fan@ki.se)

Non-alcoholic fatty liver disease (NAFLD) is a chronic liver metabolic disorder which affects up to 30% of the adult population[1]. The severity of NAFLD ranges from simple steatosis to more severe stages of non-alcoholic steatohepatitis (NASH), characterized by liver inflammation, cell ballooning, and apoptosis[1,2]. Multiple factors such as lipotoxicity, insulin resistance, and inflammation act in parallel to trigger the disease development[1–3]. Progress in understanding the role of promoting factors during disease development has revealed putative therapeutic targets including transcription factors (TFs), such as farnesoid X receptors (FXRs)[4], liver X receptors (LXRs)[5], thyroid hormone receptor (TR) β[6], and peroxisome proliferator-activated receptors (PPARs)[7]. Further targets are lipid-modulating and glucose-modulating enzymes, such as diglyceride acyltransferase (DGAT)[8], Acetyl-CoA carboxylase (ACC)[9], fatty acid synthase (FASN)[10], AMP-activated protein kinase (AMPK)[11], and metabolic hormones, such as incretins[12] and fibroblast growth factors (FGFs)[13,14]. Although gene expression (transcriptome) analysis during NAFLD progression[15,16] has identified a variety of differentially expressed marker genes, it has remained difficult to demonstrate whether differential expression is also the cause of disease progression in humans. NASH therapy remains challenging, in part due to the lack of understanding of the underlying molecular events that control those changes[1–3].

Genetically modified mice and genome-wide sequencing approaches have revealed an intrinsic relationship between transcriptomes (gene expression patterns) and epigenomes (chromatin modifications) that is directly linked to the action of diverse coregulators[2,17–19]. In contrast to well-studied TFs, the (patho-)physiological role of coregulators and epigenome alterations for the development of fatty liver disease remains to be explored. Notably, alterations in the expression and function of the histone deacetylase 3 (HDAC3) corepressor complex in macrophages and adipocytes have been linked to metabolic-inflammatory diseases, such as obesity and type 2 diabetes in humans[20–22]. In addition to HDAC3, structural core subunits of the complex include nuclear receptor corepressor (NCOR) (also NCOR1, N-CoR), silencing mediator of retinoid and TRs (SMRT, also NCOR2), and G protein pathway suppressor 2 (GPS2), all three of which have been originally identified as nuclear receptor-binding proteins, and the transducing beta-like proteins TBL1 and TBLR1 serving as coregulator exchange factors[21,23–25]. Previous studies have demonstrated that removal of NCOR, SMRT, and HDAC3 from hepatocytes in knockout (KO) mice results in increased steatosis due to disturbed lipid metabolic pathways and circadian signaling governed by TRs[26,27], LXRs[28], and Rev-Erbs[29]. PPAR-dependent genes were increased in the *Ncor* and *Smrt* KO models, supporting a role of these subunits in PPAR repression consistent with additional studies[28,30,31]. However, increased fatty acid oxidation upon removal of these subunits was not sufficient to reverse the steatosis phenotype driven by other nuclear receptor-dependent pathways including lipogenesis. One explanation could be that lipogenesis is necessary for the generation of endogenous PPARα ligands[32]. Surprisingly, TBL1 deficiency in hepatocytes resulted in reduced fatty acid oxidation, pointing at a PPARα-activating function of TBL1 independent of the corepressor complex[33]. The specific KO phenotypes of different subunits of what is thought to be the same corepressor complex suggests multiple target TFs to be affected, resulting in the modulation of in part opposing metabolic liver pathways, such as fatty acid oxidation versus lipogenesis. These data also raise the intriguing possibility that metabolic liver pathways that protect against NAFLD/NASH are controlled by a different corepressor subunit or sub-complex, the identity of which remains to be explored.

In this study, we discover a role of the GPS2 subunit in the progression of NAFLD/NASH in both mice and humans by combining the study of liver-specific *Gps2* KO (LKO) mice with correlative analysis of human transcriptome datasets[15,16]. Through diet-induced mouse models of fatty liver disease along with next-generation sequencing analysis we provide evidence that *Gps2* ablation improved liver steatosis and fibrosis, which correlated with the selective activation of lipid catabolic genes. By analyzing cistrome (GPS2 ChIP-seq), epigenome (H3K27ac, H3K4me3 ChIP-seq), and transcriptome (RNA-seq), we identified PPARα as a direct GPS2 target TF and verified the interplay of the two factors in single and double KO mice. Our analysis further reveals that GPS2 in hepatocytes functionally cooperates with NCOR but not with SMRT to repress PPARα. As these data collectively suggest that GPS2 promotes the progression of fatty liver disease via antagonizing PPARα, the selective therapeutic modulation of GPS2–PPARα interactions could be of interest for future disease interventions.

## Results

**Hepatocyte *Gps2* depletion improves MCD-induced fibrosis in mice.** We first explored the function of GPS2 in vivo using hepatocyte-specific *Gps2* LKO mice (Fig. 1a). QPCR and western blot was performed to ensure the tissue-specific KO of *Gps2* (Supplementary Fig. 1a–c). *Gps2* LKO mice started to show reduced body weight comparing with WT mice at the age of 14 months old (Supplementary Fig. 1d). *Gps2* LKO triglyceride in the very low-density lipoproteins (VLDL) fraction presented more than 50% reduction compared to WT mice, along with the total serum triglyceride level (Supplementary Fig. 1e). Lipoprotein triglyceride metabolism is tightly regulated via hepatic production and hydroxylation, but we found that serum (LPL) and hepatic lipoprotein lipase (HL) activities (Supplementary Fig. 1f) were similar between the two groups, suggesting decreased hepatic VLDL production rather than increased lipase activity. Intriguingly, *Gps2* LKO mice showed more than 50% increase in ketone body production in both fed and fasted state, suggesting a higher level of lipid oxidation in the LKO mice (Fig. 1b). This was further supported by enhanced oxygen consumption rate (OCR) in GPS2 knockdown AML12 cells both in basal and fatty acid abundant conditions (Fig. 1c). Notably, serum lipoprotein and intra-hepatic cholesterol levels were similar between WT and LKO mice (Supplementary Fig. 1g, h).

We further challenged the mice for 4 weeks with a fibrogenic methionine-deficient and choline-deficient diet (MCD). Both WT and LKO mice showed more than 30% of body weight loss yet no comparable difference between the two groups during the MCD feeding (Supplementary Fig. 1i). Liver expression of *Gps2* and *Tbl1*, but not of *Ncor* or *Smrt*, was increased upon MCD feeding for 2 and 4 weeks (Supplementary Fig. 1j). After 4 weeks MCD feeding, LKO mice showed a significant reduction of serum aspartate transaminase (AST) and alanine transaminase (ALT) activity, suggesting alleviated liver damage (Fig. 1d, e). Hematoxylin and eosin (HE) staining indicated reduced liver steatosis in LKO mice after MCD feeding (Fig. 1f), consistent with reduced liver triglyceride content (Fig. 1g). Serum lipoprotein triglyceride was not changed in MCD-fed mice (Supplementary Fig. 1k). Liver cholesterol levels were also not changed by *Gps2* ablation (Supplementary Fig. 1l). Consistently, Sirius red staining of the MCD-fed WT and LKO mice showed improved liver fibrosis in the LKO group upon MCD feeding (Fig. 1h, i), confirmed by hydroxyproline analysis (Fig. 1j).

As lipid-induced fibrosis progression is also highly related to liver inflammation[34], we treated mice with lipopolysaccharide (LPS) and tested the pro-inflammatory gene expression in the livers (Supplementary Fig. 1m). While *Il1b*, *Il6*, *Tnfα*, *Ccl2* and

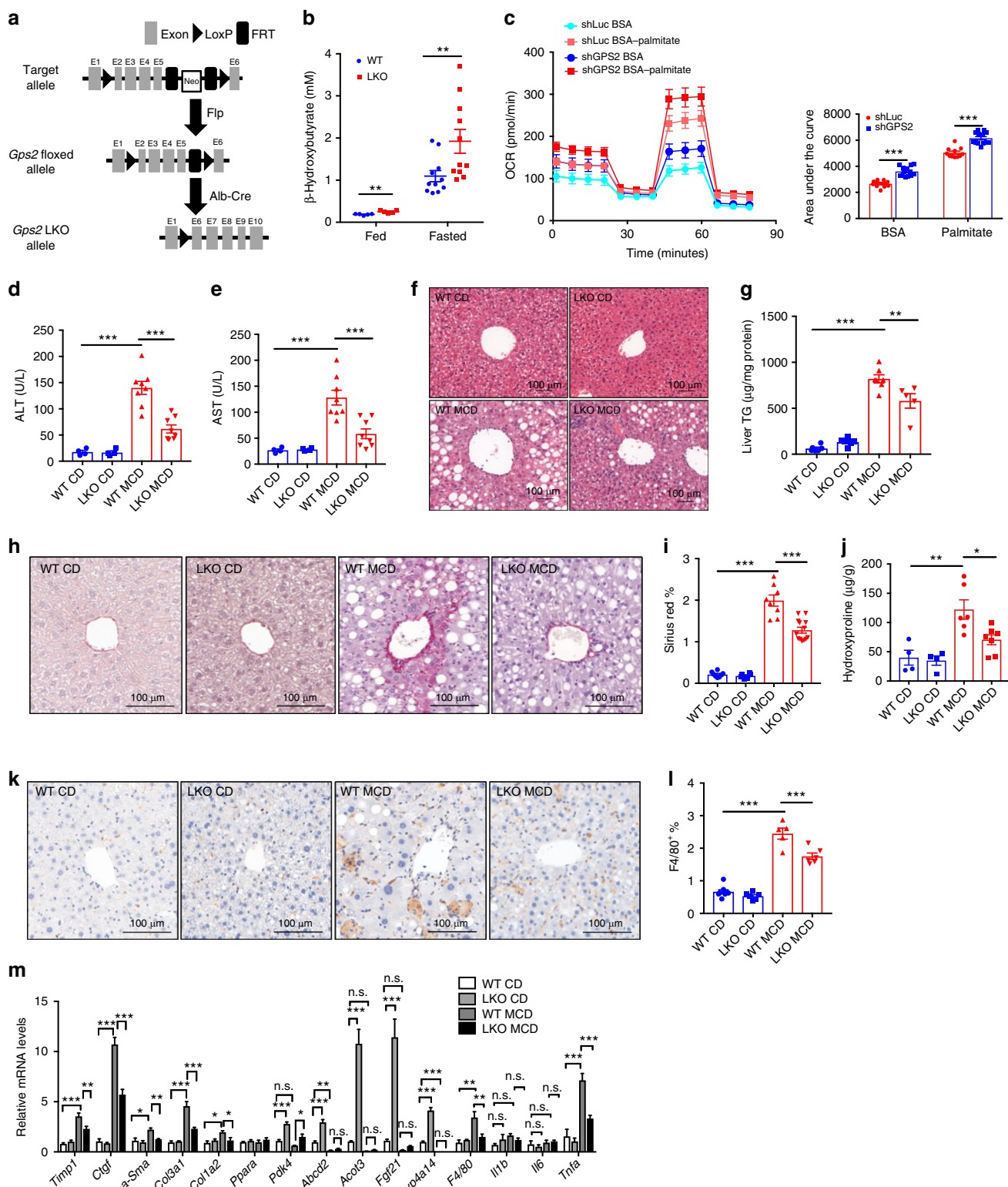

**Fig. 1** Hepatocyte *Gps2* depletion in mice improves MCD-induced fibrosis. **a** Flowchart representing the *Gps2* floxed mice generation and LKO strategy. **b** Fed and fasted ketone body analysis, fed *n* = 5, fasted *n* = 11, one-way ANOVA followed by Tukey's test. **c** Mitochondrial respiration reflected by OCR levels in BSA or BSA-palmitate treated control (shLuc) and GPS2 knockdown (shGPS2) AML12 cell lines (left panel) and area under the curve calculations (right panel), *n* = 13, one-way ANOVA followed by Tukey's test. (**d**) Serum ALT and (**e**) AST activity in chow diet (CD) and MCD-treated WT and LKO mice, *n* = 4 in CD and *n* = 8 in MCD, one-way ANOVA followed by Tukey's test. (**f**) H&E staining; (**g**) liver triglyceride, *n* = 6 in WT CD, *n* = 7 in LKO CD, *n* = 7 in WT MCD and *n* = 5 in LKO MCD; (**h**) surius red staining; (**i**) liver sirius red staining quantification, *n* = 8 in WT CD, *n* = 6 in LKO CD, *n* = 8 in WT MCD, *n* = 13 in LKO MCD; (**j**) liver hydroxyproline test, *n* = 4 in CD groups, *n* = 6 in WT MCD, *n* = 7 in LKO MCD; (**k**) F4/80 staining; (**l**) F4/80 staining quantification, *n* = 8 in WT CD, *n* = 6 in LKO CD, *n* = 5 in WT MCD, *n* = 6 in LKO MCD; and (**m**) qPCR analysis of gene expression, *n* = 5-7 in each groups, one-way ANOVA followed by Tukey's test. All data are represented as mean ± s.e.m. *$P < 0.05$, **$P < 0.01$, ***$P < 0.001$

*Ccl7* were all induced by LPS, they showed no difference between the WT and LKO groups.

However, F4/80 staining demonstrated reduced macrophage infiltration in MCD-fed LKO mice compared with WT mice (Fig. 1k, l). QPCR analysis in CD and MCD mice showed reduced fibrotic and inflammation markers, along with increased fatty acid catabolic genes (Fig. 1m).

Lastly, genes related to lipid metabolism were already up-regulated in LKO mice under basal conditions, while fibrosis markers remained unchanged (Supplementary Fig. 1n). These data collectively suggest that loss of GPS2 in hepatocytes regulates inflammation and fibrosis indirectly via regulating fatty acid metabolism.

**Gps2 LKO improves liver steatosis and insulin resistance**. We further explored the role of hepatocyte GPS2 in obesity-induced liver steatosis and insulin resistance. We first analyzed the phenotypes of mice upon chow diet feeding. Two-month-old chow diet-fed mice showed comparable blood glucose levels (Supplementary Fig. 2a), LKO mice had reduced fed and fasted (12 h fasting) blood glucose after 12 months (Supplementary Fig. 2b) and 16 months (Supplementary Fig. 2c). Fed insulin levels in both 2 months old (Supplementary Fig. 2d) and 16 months old (Supplementary Fig. 2e) mice were reduced as well, suggesting improved glucose control in LKO mice. Consistently, oral glucose tolerance test (OGTT, upon 12 h fasting) and insulin tolerance test (ITT, upon 4 h fasting) in 2 months old (Supplementary Fig. 2f, g) and 16 months old mice (Supplementary Fig. 2h, i) showed improved glucose control. We next proceeded to challenge the mice with high fat diet (HFD). Sixteen weeks of HFD increased *Gps2* expression (Supplementary Fig. 2j). We found that HFD-induced body weight gain was significantly lower in LKO mice (Fig. 2a), while food and water intake remained similar (Supplementary Fig. 2k, l). The subcutaneous fat was lower in LKO mice (Supplementary Fig. 2m). HFD-induced blood glucose levels were also significantly lower in LKO mice (Fig. 2b) although insulin levels were comparable (Fig. 2c). The OGTT and ITT test showed improved glucose control in HFD-fed LKO mice (Fig. 2d, e). Serum VLDL and total triglyceride levels (upon 4 h fasting), and HDL and total cholesterol levels were reduced in HFD LKO mice (Fig. 2f, Supplementary Fig. 2n). HE staining showed improved liver steatosis in LKO mice (Fig. 2g), confirmed by liver triglyceride analysis (Fig. 2h), although liver cholesterol level remained unchanged (Supplementary Fig. 2o). F4/80 staining showed slightly increased macrophage infiltration upon HFD treatment in WT mice (Fig. 2i, j), which was reduced in LKO mice (Fig. 2i, j). Gene expression analysis revealed reduced lipogenesis gene expression and increased fatty acid oxidation gene expression in the LKO mice liver (Fig. 2k). *Il6* and *Tnfa* were reduced in LKO mice as well (Fig. 2k). In contrast, HFD did not significantly induce fibrotic markers in the liver in both WT and LKO mice (Supplementary Fig. 2p).

Taken together, the results suggest that hepatocyte *Gps2* KO reduces body weight and plays a key role in modulating both lipid metabolism and glucose homeostasis in obese mice.

**PPARα is a direct target of GPS2 in hepatocytes**. We continued to investigate underlying mechanisms involved in the regulatory role of GPS2 in the liver. We first identified GPS2 signature genes from WT and LKO livers in chow diet fed mice using RNA-seq (Fig. 3a). KEGG pathway analysis revealed enrichment of up-regulated genes in multiple metabolic pathways with PPAR pathways appeared in top enriched signaling pathways (Fig. 3b). Given the dominant role of PPARα in liver lipid oxidation[35], we identified PPARα target genes in mouse livers using available

transcriptome datasets[36] (GSE73298 and GSE73299, Supplementary Fig. 3a), followed by a comparison of GPS2-regulated liver transcriptomes with both global (Supplementary Fig. 3b) and liver-specific *Ppara* KO mice (Supplementary Fig. 3c). We found that around 20% of the PPARα target genes were regulated by GPS2 in both groups of comparisons, and among these more than 80% were repressed by GPS2 (Supplementary Fig. 3b, c). We additionally compared the PPARα target genes concluded from a previous report[37]. Among 121 target genes, 52 (including *Cyp4a14, Pdk4, Fgf21*) were up-regulated in *Gps2* LKO livers, 27 were down-regulated and 42 remained unchanged (Supplementary Fig. 3d, e). Intriguingly, LXR agonist GW3965 treatment did not induce significant differences between WT and LKO mice (Supplementary Fig. 3f), suggesting GPS2 selectivity towards PPARα.

We next tested liver gene expression in fasted (Fig. 3c) and GW7647-treated mice (Fig. 3d). The results show that LKO mice had enhanced target gene responses to both treatments (Fig. 3c, d). We then determined the liver GPS2 cistrome using ChIP-seq. In agreement with the transcriptome data, TF binding motif analysis revealed PPAR-binding sites among the top GPS2-occupied sites (Fig. 3e). Co-immunoprecipitation assays revealed that interaction with PPARα was dependent on the GPS2 region aa 100–155 (Fig. 3f), part of the previously characterized nuclear receptor-binding domain[38].

To demonstrate the requirement of PPARα for GPS2 repression of lipid metabolic genes, we generated *Gps2* and *Ppara* double KO (PGKO) mice by cross-breeding the two single KO strains (Fig. 3g). PGKO and *Ppara* KO (PKO) mice had similar body weight (Supplementary Fig. 4a), liver weight (Supplementary Fig. 4b), liver cholesterol, and triglyceride levels (Supplementary Fig. 4c, d). Notably, qPCR showed that *Gps2* ablation did not induce the PPARα target genes *Pdk4, Cyp4a14, Fgf21* in PGKO livers that lack PPARα (Fig. 3g). Lipid oxidation seen in *Gps2* LKO versus WT mice was not observed in PGKO versus PKO mice, as indicated by unchanged fed and fasted ketone body generation (Fig. 3h), suggesting that GPS2 repression of lipid oxidation was dependent on PPARα. Blood glucose (Supplementary Fig. 4e) and insulin levels (Supplementary Fig. 4f) were also not changed between the two groups. Above all, serum lipoprotein and triglyceride levels remained significantly increased in PKO mice, but were unchanged by *Gps2* depletion (Supplementary Fig. 4g).

Collectively, these data identify PPARα as a major target TF for GPS2 in the liver.

**GPS2 requires PPARα to modulate the hepatic epigenome**. To define the GPS2-dependent cistrome and epigenome in mouse liver, we performed ChIP-seq of GPS2 along with H3K27ac and H3K4me3 in WT and LKO mice (Fig. 4a). Alignment of the ChIP-seq peak heatmaps of H3K27ac, H3K4me3, and H3K4me1 (GSE29218)[39] with GPS2 revealed the presence of GPS2 in both active promoter and enhancer regions (Fig. 4a), which was confirmed by GPS2 peak distribution analysis (Supplementary Fig. 4h). Comparison of the GPS2-dependent transcriptome and epigenome revealed that transcriptional and epigenetic activation at GPS2-sensitive gene loci were highly coordinated (Supplementary Fig. 4i–n). Both H3K27ac and H3K4me3 levels are higher in LKO up-regulated (GPS2-repressed) genes compared to unchanged (GPS2-resistant) genes. Likewise, downregulated gene loci had lower H3K27ac and H3K4me3 levels (Supplementary Fig. 4i, j). Correlation analysis of GPS2-sensitive genes showed significant correlation with both H3K27ac and H3K4me3 levels (Supplementary Fig. 4k, l), marking direct GPS2-mediated gene regulation via those promoters and enhancers (Supplementary

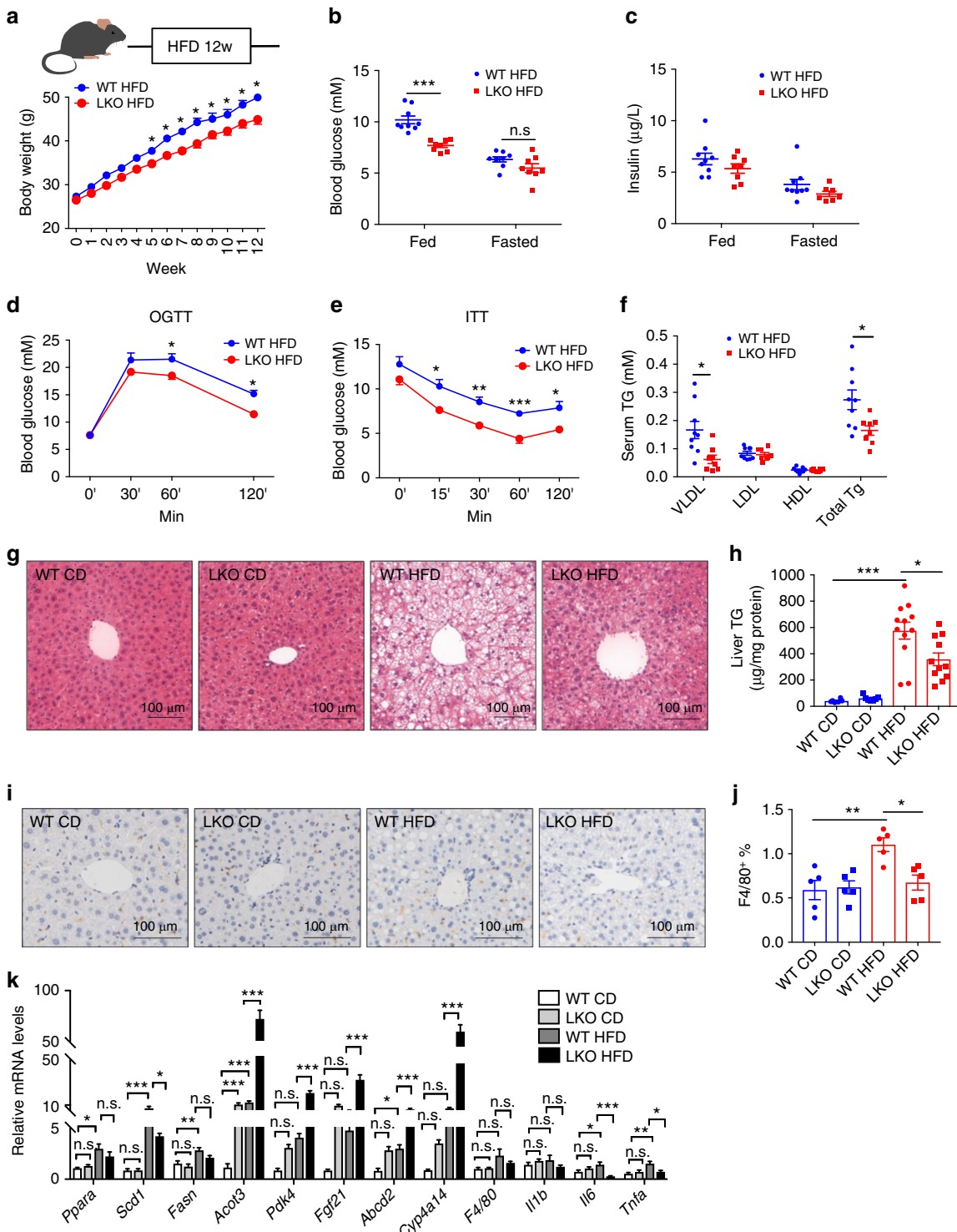

**Fig. 2** *Gps2* LKO mice are resistant to HFD. **a** Body weight change during 12 weeks HFD feeding in WT and LKO mice, $n = 9$–16 in WT and $n = 10$–17 in LKO, non-parametric Mann–Whitney test. Fed and fasted (**b**) glucose and (**c**) insulin in WT and LKO mice, $n = 9$ in WT and $n = 8$ in LKO, non-parametric Mann–Whitney test. (**d**) OGTT and (**e**) ITT in WT and LKO mice after 12 weeks of HFD, $n = 10$, non-parametric Mann–Whitney test. (**f**) Triglyceride in VLDL fractions, $n = 8$–9 in each group. non-parametric Mann–Whitney test. (**g**) H&E staining; (**h**) liver triglyceride levels, $n = 6$ in WT CD, $n = 7$ in LKO CD, $n = 12$ in WT HFD and $n = 11$ in LKO HFD, one-way ANOVA followed by Tukey's test. (**i**) F4/80 staining and (**j**) quantification, $n = 5$ in each group. (**k**) QPCR analysis of liver mRNA expressions, $n = 5$–9 in each group from 12 weeks CD-fed or HFD-fed WT and LKO mice, one-way ANOVA followed by Tukey's test. All data are represented as mean ± s.e.m. $*P < 0.05$, $**P < 0.01$, $***P < 0.001$

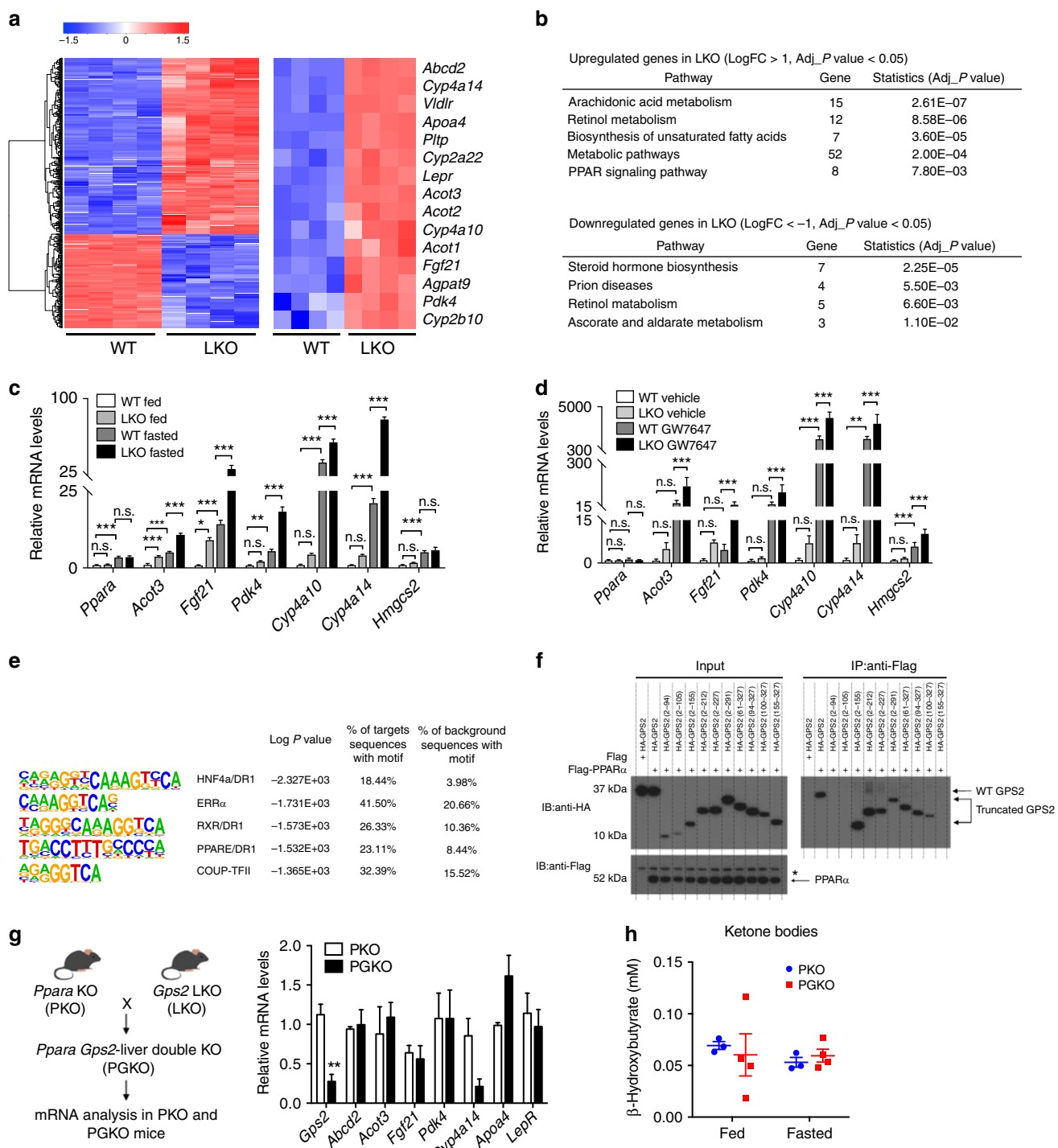

**Fig. 3** GPS2 depletion causes PPARα activation. **a** Heatmap representing the top 1000 significant (sorted based on adj *P* value) gene expression in CD-fed WT and LKO livers. **b** KEGG pathway analysis of significantly up-regulated and down-regulated genes in LKO mice. **c**, **d** QPCR analysis of mRNA expression in WT and LKO mice upon (**c**) fasting, *n* = 5 in each group; and (**d**) PPARα agonist (GW7647) treatment, *n* = 5 in the vehicle groups, *n* = 7 in the GW7647-treated groups, one-way ANOVA followed by Tukey's test. **e** Motif analysis of GPS2-occupied regions in mouse liver. **f** Co-immunoprecipitation of GPS2 domains with PPARα in 293 cells (* represents non-specific bands). **g** QPCR analysis of GPS2-regulated genes and (**h**) fed and fasted serum ketone bodies in PPARα KO (PKO) and liver *Gps2/Ppara* double KO (PGKO) mice livers, *n* = 3 in PKO and *n* = 4 in PGKO, non-parametric Mann–Whitney test. All data are represented as mean ± s.e.m. *$P < 0.05$, **$P < 0.01$, ***$P < 0.001$

Fig. 4m, n), as represented for repression in the *Pdk4* and *Cyp4a14* loci (Fig. 4b, c).

We further compared the ChIP-seq profiles of GPS2 and PPARα in mouse livers. GPS2 and PPARα were co-localized in

the *Pdk4* and *Cyp4a14* loci (Fig. 4d, e, GSE61817)[40]. Globally, around 85% of the GPS2 peaks were overlapped with PPARα-binding sites (Fig. 4f). Moreover, GPS2 ablation increased H3K27ac in *Pdk4* and *Cyp4a14* promoter/enhancer loci, which

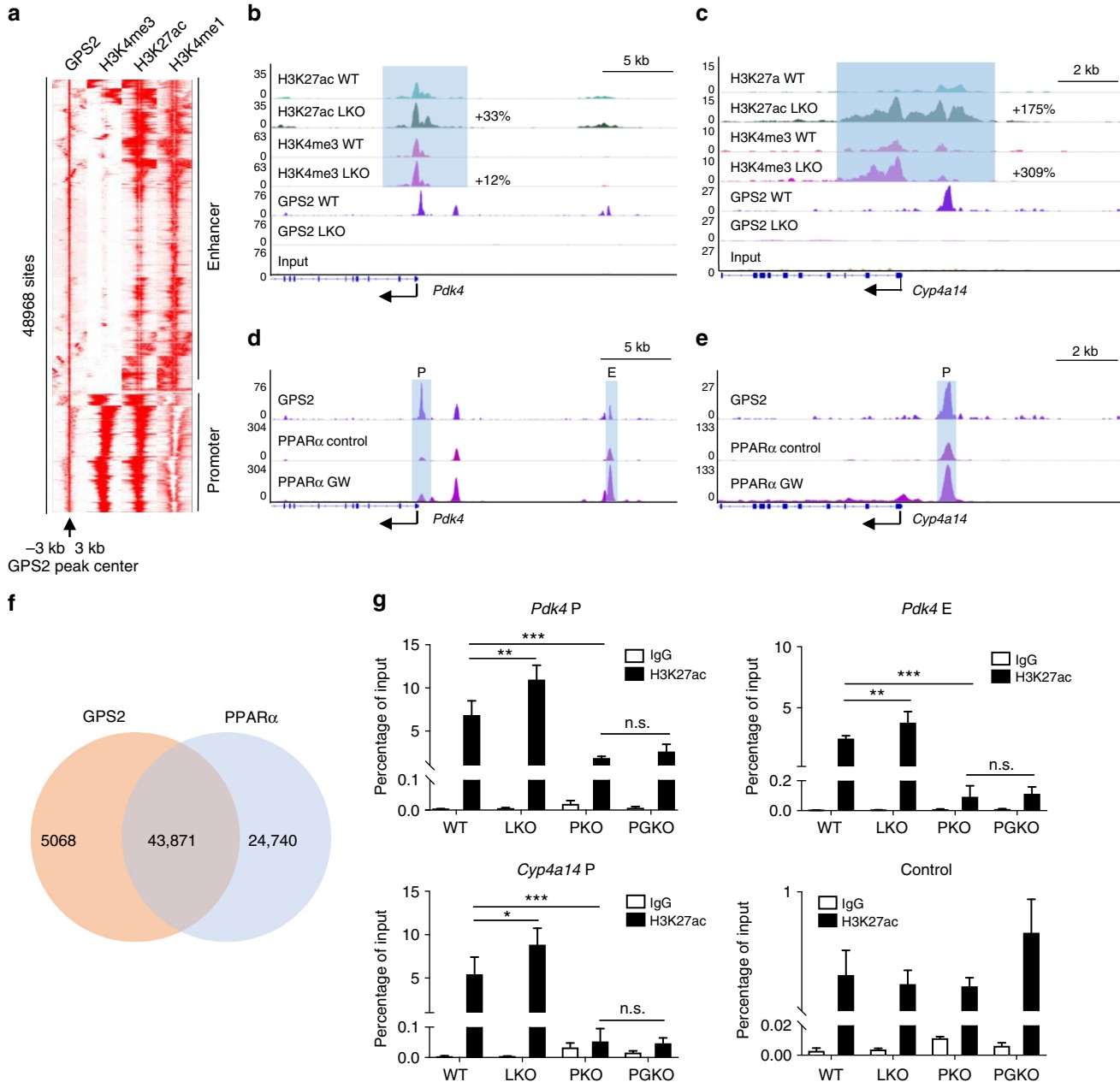

**Fig. 4** GPS2 represses PPARα-regulated promoters and enhancers. **a** GPS2, H3K4me3, H3K27ac, and H3K4me1 ChIP-seq peaks are aligned to the GPS2 peak center in a window of ±3 kb. **b**, **c** ChIP-seq tracks of GPS2, H3K27ac, and H3K4me3 peaks at (**b**) *Pdk4* and (**c**) *Cyp4a14* loci, percentage of increase (LKO versus WT) represents logFC of each peak and is calculated from biological duplicates. **d**, **e** ChIP-seq tracks of GPS2, PPARα (vehicle and GW7647 treated, GSE61817) peaks at (**d**) *Pdk4* and (**e**) *Cyp4a14* loci. **f** Venn diagram representing the overlapped GPS2 and PPARα peaks detected at least twice in mouse liver. **g** Further ChIP qPCR validation of H3K27ac at *Pdk4* promoter (upper left), *Pdk4* enhancer (upper right), *Cyp4a14* promoter (lower left) and control (lower right) locus in WT, LKO, PKO, and PGKO livers, $n = 5$ in WT and LKO groups, $n = 4$ in PKO group and $n = 3$ in PGKO group, one-way ANOVA followed by Tukey's test. All data are represented as mean ± s.e.m.*$P < 0.05$, **$P < 0.01$, ***$P < 0.001$

was not observed in PGKO versus PKO mice (Fig. 4g), suggesting that PPARα is required for GPS2-mediated epigenomic repression at those gene loci.

**GPS2 cooperates with NCOR in hepatocytes**. The apparent PPARα-selectivity of GPS2 repression was surprising as it was not seen in the comparable liver KO models for other complex subunits. Thus, we next investigated whether GPS2 functions within the corepressor complex to modulate liver gene expression. We first compared the chromatin recruitment of NCOR and SMRT

with GPS2 in mouse livers. Cistrome analysis revealed that all three subunits shared more than 50% of all binding sites in the liver genome (Supplementary Fig. 5a, b, GSE26345 and GSE51045)[29,41]. *Gps2* LKO livers shared overlapping transcriptome signatures with both *Ncor* and *Smrt* LKO livers[26,41] (Supplementary Fig. 5c, d). However, KEGG pathway analysis of the genes co-repressed by GPS2 and NCOR revealed PPAR signaling in the top list, along with liver metabolic pathways (Supplementary Fig. 5e), which was not observed for the genes co-repressed by GPS2 and SMRT (Supplementary Fig. 5f). At the epigenome level, H3K27ac levels were similarly increased in *Gps2*

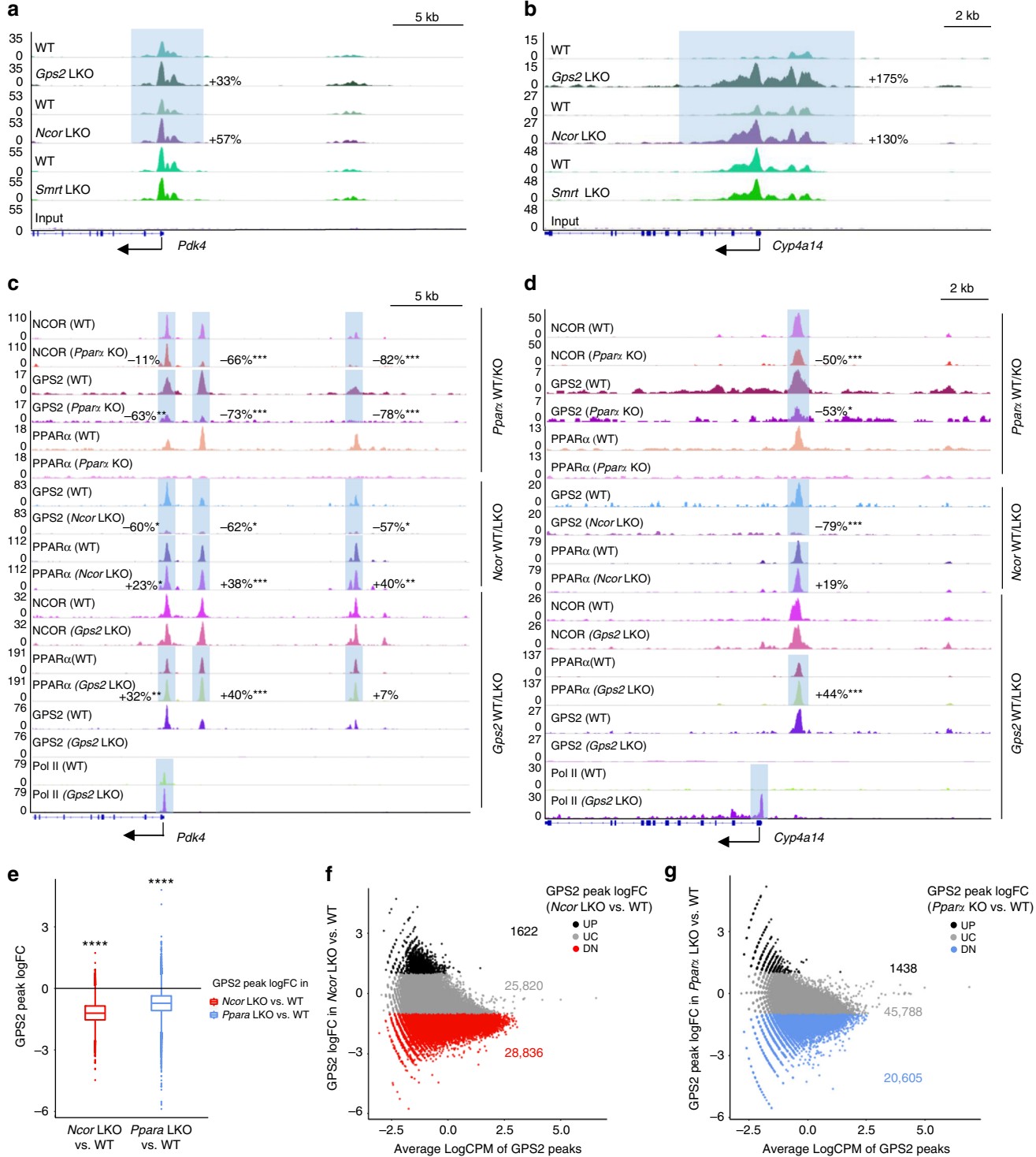

**Fig. 5** GPS2 cooperates with NCOR to repress target gene expression. **a**, **b** ChIP-seq tracks representing H3K27ac at (**a**) *Pdk4* and (**b**) *Cyp4a14* loci in *Gps2*, *Ncor* and *Smrt* LKO and respective WT mouse livers. Percentage of increase represents logFC of each peak and is calculated from biological duplicates (for *Gps2* LKO versus WT) or triplicates (for *Ncor* or *Smrt* LKO versus WT). **c**, **d** ChIP-seq tracks representing NCOR, GPS2, PPARα, and Pol II recruitment in *Pparα* KO, *Ncor* LKO, *Gps2* LKO and respective WT livers at (**c**) *Pdk4* and (**d**) *Cyp4a14* loci. Percentage of increase represents logFC of each peak and is calculated from biological triplicates. **e** Boxplot representing average logFC of co-localized GPS2 peaks in *Ncor* LKO and *Pparα* KO versus WT livers ($n = 3$ in each group). **f**, **g** MA plots showing the average logFC of GPS2 peaks between (**f**) *Ncor* LKO versus WT; and (**g**) *Pparα* KO versus WT livers, against average logCPM. UP, UN, and DN represents upregulated, unchanged, and downregulated peaks, respectively. All data are represented as mean ± s.e.m. *$P < 0.05$, **$P < 0.01$, ***$P < 0.001$

LKO and in *Ncor* LKO livers, but not in *Smrt* LKO livers, at the *Pdk4* and *Cyp4a14* loci (Fig. 5a, b). Intriguingly, binding of GPS2 and NCOR was reduced in *Pparα* KO livers at the *Pdk4* and *Cyp4a14* loci (Fig. 5c, d, Supplementary Fig. 5g) while SMRT binding was not affected (Supplementary Fig. 5h). In addition, GPS2 recruitment to these loci relied on NCOR but not on SMRT and was abolished in *Ncor* LKO livers (Fig. 5c, d, Supplementary Fig. 5g, i). The recruitment changes were not limited to above genes since global analysis of GPS2 ChIP-seq revealed reduced average binding in GPS2/NCOR or GPS2/PPARα co-localized loci in *Ncor* LKO or *Pparα* KO versus WT livers, respectively (Fig. 5e). Consistently, there were far more GPS2 peaks that were reduced comparing with the increased ones in *Ncor* LKO (Fig. 5f) or *Pparα* KO (Fig. 5g) versus WT livers. Similar analysis of PPARα peaks in *Gps2* and *Ncor* LKO versus WT liver showed increased average PPARα recruitment (Supplementary Fig. 5j). The MA plot also revealed more enhanced PPARα binding in both *Ncor* and *Gps2* LKO livers (Supplementary Fig. 5k, l). In addition, PPARα recruitment was slightly increased at NCOR-repressed gene loci (i.e. upregulated in *Ncor* LKO versus WT mice), including the *Pdk4* and *Cyp4a14* loci (Fig. 5c, d, Supplementary Fig. 6a), but showed significant increase at GPS2-repressed gene loci (Fig. 5c, d, Supplementary Fig. 6b). In contrast, NCOR, SMRT, and HDAC3 recruitment was only modestly changed in *Gps2* LKO livers (Fig. 5c, d, Supplementary Fig. 6c).

The interactions of GPS2, NCOR, and PPARα were further investigated in vitro using co-immunoprecipitations. We found that full-length GPS2 enhanced the PPARα interaction with NCOR (Supplementary Fig. 6d), while this was not seen with truncated GPS2 variants removing the NCOR-interaction domain (aa 61–94) or the PPARα-interaction domain (aa 100–155). These data support that the loss of GPS2 in hepatocytes de-stabilizes interactions of PPARα with the NCOR complex, and that release of the complex potentially facilitates the binding of RNA polymerase (Pol II) (Fig. 5c, d).

In sum, the above data suggest that GPS2 repression of PPARα target genes in hepatocytes involves direct interactions of PPARα with GPS2 and NCOR, which serves as the primary docking site of the corepressor complex to chromatin in hepatocytes.

**Liver *GPS2* expression correlates with NASH and fibrosis in humans**. To determine the clinical relevance of GPS2 in human disease progression, we compared gene expression levels in both NASH (Fig. 6 and Supplementary Fig. 6e–j) and NAFLD (Fig. 6 and Supplementary Fig. 6e–j) human subjects from published transcriptome datasets (GSE83452 and GSE49541)[15,16]. The comparative analysis performed in a cohort of 44 control and 104 NASH liver samples identified 193 upregulated and 58 down-regulated NASH signature genes as concluded in the previous publication (Fig. 6a). Expression of *GPS2* was not different in different stages of NASH liver (Supplementary Fig. 6e). However, correlative analysis in the 104 NASH subjects showed a total of 74 genes significantly associated with *GPS2* mRNA expression and 66 were positively correlated (Fig. 6b, Supplementary Table 2). Among them were fibrogenic genes including *TGFB*, *TIMP1* (Supplementary Fig. 6f, g), and the inflammation marker *CD68* (Supplementary Fig. 6h). Analysis of the other complex subunits revealed associations of these signature genes with *NCOR*, *TBL1*, *TBLR1*, and *HDAC3*, but notably not with *SMRT* (Fig. 6b). To further validate these results, we analyzed another transcriptome dataset (GSE49541) which contained 73 NAFLD patients at different fibrosis stages. *GPS2* expression was positively correlated with the fibrogenic gene expression, including *TGFB* and *TIMP1* (Fig. 6c, d).

Moreover, *GPS2* expression was higher in NASH fibrosis compared to non-fibrosis liver biopsies (Fig. 6e) and was restored after weight loss in paired obese human subjects after dietary intervention or gastric bypass surgery (GABY) (Fig. 6f). The *GPS2* mRNA was also positively correlated with HbA1c (Supplementary Fig. 6i). Intriguingly, correlative analysis of *GPS2* and *PPARA* expression with the 72 GPS2-sensitive and NASH-related genes in the dataset from 104 human NASH patients showed an overall opposite correlation (Fig. 6g), while *GPS2* and *PPARA* expression was not correlated (Supplementary Fig. 6j).

We conclude from above data that liver GPS2 expression and function could be causatively correlated with the progression of NAFLD towards NASH via regulating PPARα-coupled liver lipid metabolism (Fig. 6h).

## Discussion

Our study uncovers a previously unknown role of GPS2 as an epigenetic modulator which inhibits liver lipid catabolism and thereby promotes the development of NALFD. *Gps2* LKO caused the activation of promoters and enhancers controlling fatty acid oxidation genes in hepatocytes, which was dependent on PPARα activation. As a result of loss of PPARα repression, *Gps2* LKO mice showed alleviated liver steatosis upon HFD feeding and improved fibrosis upon MCD feeding, due to increased lipid burning as detected by elevated ketone body levels. Remarkably, the protective phenotype of the *Gps2* LKO mice is unique amongst hitherto described coregulator KO mouse models in the context of NAFLD as it is the only model which improved diet-induced fatty liver disease instead of worsening it (for references, see Introduction).

This hepatic function of GPS2 appears to be conserved between mice and humans as *GPS2* levels correlate with fibrogenic and inflammatory gene expression in human NAFLD/NASH livers. Our study might thus provide hepatocyte-based epigenetic explanations for the diverse susceptibility in NAFLD/NASH patients to develop more severe stages of liver fibrosis and ultimately liver cancer, in addition to alterations in other cell types such as liver-resident immune cells[1].

Previous studies in adipocytes[20,42] and macrophages[21] support a key role of GPS2 in repressing inflammation, and loss of GPS2 sensitized these cell-types to develop metabolic stress-induced inflammation (metaflammation) and insulin resistance, as shown for adipose tissue[21]. Importantly, this was not observed in the *Gps2* LKO mice. Gene expression of representative pro-inflammatory genes was not significantly different between WT and LKO livers, and injection of LPS led to similar responses, as shown for *Ccl2* and *Ccl7*, major GPS2 target genes in adipocytes and macrophages. These results are consistent with previous data demonstrating that in human hepatocytes GPS2 depletion had no effect on the inflammatory acute phase response but was essential for anti-inflammatory trans-repression of this response by nuclear receptors[43].

Our earlier studies in human hepatocyte cell lines suggested a role of GPS2 in cholesterol metabolism which may influence the NAFLD phenotype[38,44]. However, chow diet-fed *Gps2* LKO versus WT mice did not show differences in both serum and hepatic cholesterol levels despite triglyceride variation. Caution should be exercised in translating this aspect of the mouse phenotype to humans, as they differ in lipoprotein metabolism from mice due to absence of cholesterol ester transfer protein (CETP)[45]. There are also human–mouse differences in the regulation of cholesterol 7α-hydroxylase (CYP7A1), the rate-limiting enzyme in the classic bile acid synthetic pathway, by the nuclear receptors LXR, FXR, and SHP. While this pathway appears to be regulated by GPS2 in

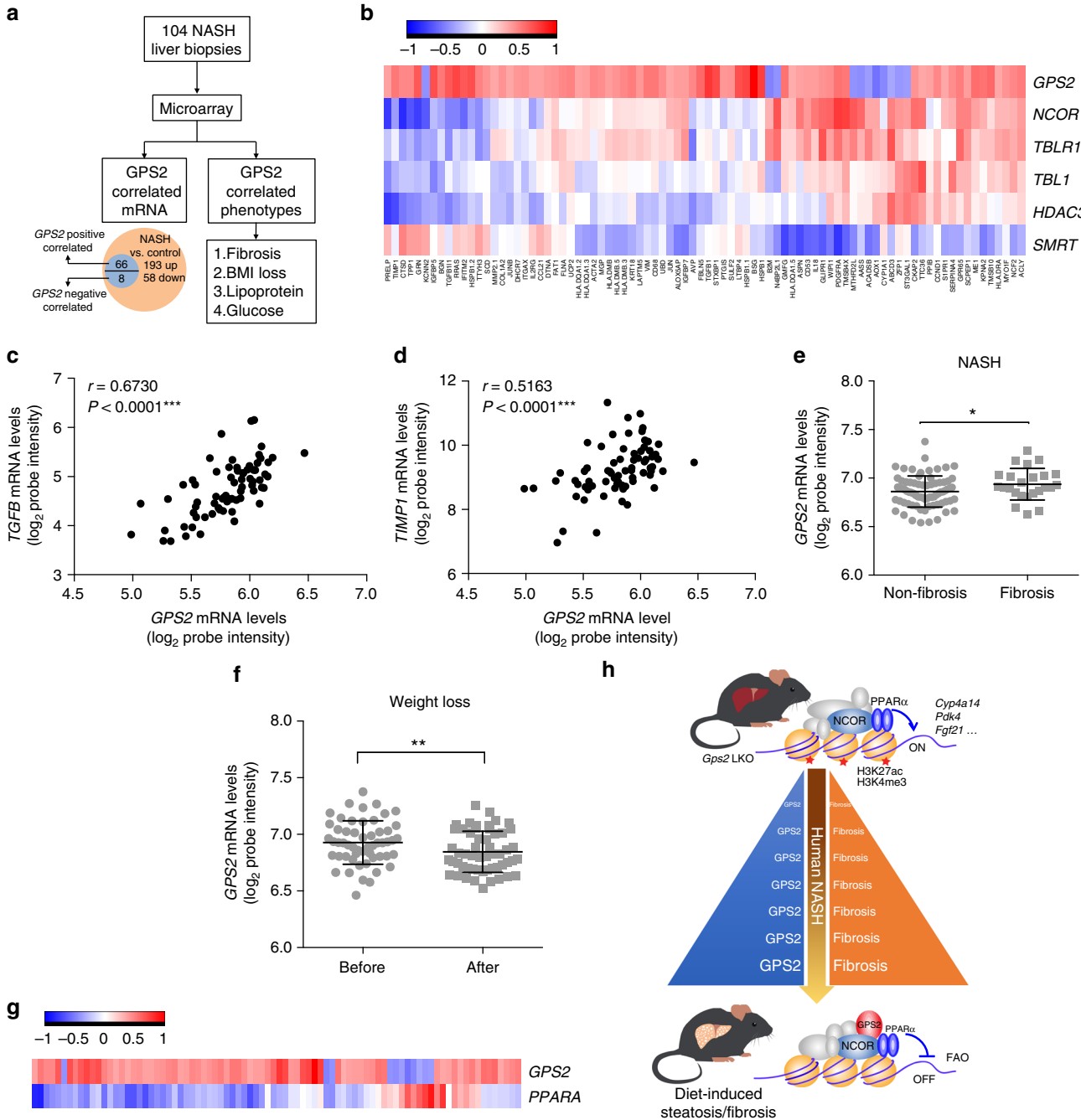

**Fig. 6** *GPS2* expression associates with NASH and fibrosis in human. **a** Flowchart representing the analysis of microarray results in the human liver biopsies. **b** Heatmap representing the rho correlation coefficient of *GPS2* and other core subunits with NASH-regulated genes in human NASH liver biopsies. **c**, **d** Correlation analysis of *GPS2* mRNA level with (**c**) *TGFB* and (**d**) *TIMP1* in human NAFLD liver samples (GSE49541), $n = 72$, non-parametric Spearman's test. **e** *GPS2* mRNA level in non-fibrosis and fibrosis liver biopsies in the NASH subjects, $n$(NASH non-fibrosis) = 74, $n$(NASH fibrosis) = 24, unpaired nonparametric Mann–Whitney test. **f** *GPS2* mRNA level in NASH after weight loss in paired obese subjects, $n = 54$, paired nonparametric Wilcoxon matched-pairs signed rank test. **g** Heatmap representing the rho correlation coefficient of *GPS2* and *PPARA* with NASH-regulated genes in human NASH liver biospies ($n = 104$). **h** Model illustrating the causal relationship between GPS2 expression and PPARα function in hepatocytes in the context of fatty liver disease. All data are represented as mean ± s.e.m. *$P < 0.05$, **$P < 0.01$, ***$P < 0.001$

human hepatocytes[44], the activation of key LXR target genes by agonist treatment was similar in WT versus LKO mice, suggesting that liver LXR pathways in mice are not controlled by GPS2.

Despite the evidence that GPS2 likely functions as a core subunit of the NCOR/SMRT/HDAC3 corepressor complex in many metabolic tissues and cell types[21,26,38,43,44], it may seem puzzling that the KO phenotypes for individual subunits were quite distinct with only partial overlap in specific pathways. We

show here that in mouse hepatocytes GPS2 functionally cooperates with NCOR to repress PPARα-dependent pathways. This is in contrast to the situation in adipocytes and macrophages where GPS2 functionally cooperates with SMRT to repress proinflammatory pathways[20,21]. While the molecular reasons for this cell-type selective partnership remain enigmatic, they are not simply related to the differential expression and/or chromatin-binding features of NCOR and SMRT. Indeed, our liver cistrome

analysis reveals that the core subunits GPS2, NCOR, SMRT, and HDAC3 co-occupy the majority of promoters and enhancers of GPS2-regulated genes, including PPARα-targets linked to fatty acid oxidation and to the LKO phenotype. This confirms our previous notion made in macrophages that the entire complex seems to be present at chromatin elements, yet this is not sufficient to predict function in the regulation of adjacent genes. These data collectively validate our recent concept of functionally distinct corepressor sub-complexes[19].

Our new work identifies with GPS2 an additional core subunit of the complex that also interacts with PPARα and contributes to the overall repression in vivo. The Co-IP data and the inclusion of GPS2 mutants add in vitro support that GPS2 utilizes different domains to bind PPARα vs. NCOR, and that this binding may be cooperative to increase PPARα interactions with NCOR. This is fully compatible with a model where cooperative interactions of two subunits explain why depletion of either NCOR or GPS2 reduces PPARα repression capacity and target gene expression in vivo.

There is evidence that GPS2 forms a stable core complex with NCOR/SMRT and TBL/TBLR1, supported by NMR structure data[23] and IP-mass spec data[24,46]. NCOR has been previously known to be the main receptor (also PPARα)-binding subunit of the complex, and our new data do not challenge this view. Indeed, our ChIP-seq data confirm that most GPS2 interactions with PPARα at chromatin are lost in Ncor KO hepatocytes, while NCOR interactions were not abolished in Gps2 KO hepatocytes. These results indicate that NCOR does not require GPS2 to interact with PPARα in vivo, thus contrasting the GPS2–NCOR bridging model suggested for agonist-bound PPARγ[47].

However, there is so far no direct evidence whether this corepressor complex core forms stable ternary complexes with nuclear receptors, including PPARα. The current data rather suggest that repression in vivo results from dynamic transient interactions of the corepressor complex with TF targets[22]. NCOR and GPS2 are proposed to be largely intrinsically disordered in nature, which may reflect the involvement in many specific but relatively low-affinity interactions[48].

Regarding the details of the PPARα interactions with GPS2 and NCOR, there is ample evidence suggesting that both proteins interact with distinct surfaces at the ligand-binding domain (LBD). In case of NCOR, the receptor interaction domain (RID) is largely unstructured but adopts upon interaction a helical peptide structure (the CORNR box) that interacts with the conserved cofactor surface at the LBD helices 3–5 (but excludes AF-2 helix 12), see for example, the structure of the PPARα LBD with a corepressor peptide in the presence of an antagonist[30,49,50]. Although there is no structure data for any RID of GPS2, in vitro assays using receptor mutations and NCOR peptide competition strongly suggest that GPS2 binds to a surface that is different from the AF-2/NCOR-binding surface. In particular, the GPS2 surface seems exposed irrespective of the ligand status, marking a fundamental difference to the classic coactivator (LXXLL motif) corepressor (CORNR motif) surfaces[38,47]. One interesting in vivo scenario is that activation by endogenous PPARα agonists may lead to selective disruption of NCOR interactions, while GPS2 interactions may serve to stabilize the corepressor complex interactions, contributing to the GPS2 KO phenotype described here.

Remarkably, the functional cooperation of GPS2 with NCOR and its antagonism of PPARα is further supported by the correlation analysis of human NAFLD/NASH liver transcriptomes, suggesting conservation of the GPS2–NCOR partnership in driving NAFLD in mice and humans. The proposed link of GPS2 to NCOR in hepatocytes raises questions as to the different corresponding KO phenotypes. While Gps2 LKO resulted in

improved liver steatosis and reduced lipid accumulation, Ncor LKO resulted in the opposite[26,28]. A likely explanation could be the partially distinct nuclear receptor and target gene selectivity of each subunit, with NCOR but not GPS2 repressing also LXRα in addition to PPARα[28]. As a consequence, lipogenic pathways were only up-regulated in Ncor LKO mice and likely contributed to the worsened fatty liver phenotype. Interestingly, the NCOR study provides some clues for how corepressor complex function, and in particular the subunit cooperation, might be further regulated to specify transcriptional outcomes. Nuclear receptor-selectivity seems regulated by insulin/AKT-mediated S1460 phosphorylation of NCOR, resulting in de-repression of LXR-dependent lipogenesis in the fed condition and PPARα-regulated fatty acid oxidation in the fasting condition [28].

The transcriptome comparison of Gps2 and Ncor LKO mice supported the different phenotypes, in particular the PPARα-selective cooperation of NCOR with GPS2. Gene ontology analysis indicated that PPAR signaling was amongst the top pathways of the 30% of genes co-regulated by NCOR and GPS2, while LXR targets genes linked to lipogenesis were not co-regulated. Moreover, our analysis revealed interesting differences regarding the regulation of Fgf21, a major PPARα target gene in liver. Gps2 LKO resulted in an eight-fold increase in Fgf21 expression in the chow diet condition which was PPARα-dependent, the induction level almost equivalent to fasting effects, while in Ncor and Smrt LKO mice Fgf21 de-repression was not observed. The encoded FGF21 is a master regulator of metabolism which affects multiple pathways including fatty acid oxidation, and administration to mice has been shown to improve NAFLD[13,51,52]. The reason why Fgf21 is selectively up-regulated in Gps2 but not in Ncor or Smrt LKO mice is unknown, but there is one report suggesting the HDAC3 subunit of the complex to be involved[53]. The study shows that pharmacological HDAC3 inhibition led to elevated Fgf21 expression in mouse and human liver cell lines. In addition to FGF21, most of the pathways and TFs (e.g. Rev-Erbs, Prox1) affected in Hdac3 LKO mice[29,46] were not affected in Gps2 LKO mice, consistent with the broader target range of HDAC3 along with its intrinsic de-acetylation activity.

The unique phenotype of Gps2 LKO mice supports the idea that the GPS2 target range in vivo is limited, as compared to NCOR, SMRT, and HDAC3 and to other coregulators expressed in the liver[17]. This implies that the manipulation of GPS2 function could be more selective than the manipulation of other coregulators. In conclusion, our study identifies GPS2 as an epigenome modifier and PPARα-selective corepressor in hepatocytes whose inhibition has the therapeutic potential to reverse the progression of NASH toward fibrosis.

## Methods

**Patients**. The patient clinical information and human liver biopsy transcriptome data are from a previously published Belgium cohort[15] collected from overweight individuals visiting the Obesity Clinic at the Antwerp University Hospital. The patient information and exclusion criteria were all described previously[15]. Briefly, the cohort used in this study is composed of 104 NASH patients with paired liver biopsies of 35 samples after dietary intervention and 39 samples after GABY (combined as weight loss group). The fibrosis stage was determined by pathological analysis of the liver biopsies. BMI (loss), HbA1c, HDL-c, and LDL-c were determined as described previously[15], and the study was approved by the Ethical Committee of the Antwerp University Hospital (file 6/25/125).

**Animals**. Gps2[flox/flox] mice were generated at Ozgene Pty, Ltd. (Bentley DC, Australia) as previously described[21]. To generate LKO mice, Gps2[flox/flox] mice were crossed with Alb-Cre mice (B6.Cg-Speer6-ps1[Tg(Alb-cre)21Mgn]/J; Jackson Laboratory stock no. 003574). Gps2[flox/flox] mice from the same breedings were used as control (labeled as WT).

Pparα-deficient C57Bl/6J mice[54] (B6;129S4-Pparα[tm1Gonz]/J; Stock no. 008154) were obtained from Jackson Laboratory.

Liver-specific Gps2 and Ppara double KO mice were generated by breeding the liver Gps2 KO mice with the Ppara-deficient ($Gps2^{\text{flox/flox}}$ Alb-$Cre^{+/-}Ppara^{-/-}$) mice. $Gps2^{\text{flox/flox}}$ Alb-$Cre^{-/-}Ppara^{-/-}$ mice were used as control.

Liver-specific Ncor and Smrt KO mice were generated as previously described[26].

All animals were randomly assigned to each experimental group. The investigator was not blinded to the experimental groups during the study.

All animal experiments were approved by the respective national ethical boards (Swedish Board of Agriculture, Stockholm South, S28-12, S30-14, S135-12, S29-14, ID907) and conducted in accordance with the guidelines stated in the International Guiding Principles for Biomedical Research Involving Animals, developed by the Council for International Organizations of Medical Sciences (CIOMS). All mice strains were bred and maintained at the Center for Comparative Medicine at Karolinska Institutet and University Hospital (PKL, Huddinge, Sweden).

**Cell culture studies.** HEK293 (ATCC, CRL-1573) and AML12 (ATCC, CRL-2254) cells were tested for mycoplasma contamination. AML12 cells were authenticated by testing liver albumin expression before experiments. HEK293 were cultured in DMEM supplemented with 10% FBS and 1% antibiotics, and AML12 were cultured in DMEM/F12 high glucose supplemented with 10% FBS, 1% antibiotics, 1% ITS (10 µg/ml insulin, 5.5 µg/ml transferrin, and 5 ng/ml selenium), and 40 ng/ml dexamethasone. Both cells were maintained at 37 °C and 5% $CO_2$.

**Seahorse analysis of OCR in AML12 cells.** The Cell Mito Stress Test was performed according to the provided protocol. AML12 cells were seeded to the Seahorse XFe96 Cell Culture Microplate for 24 h before transduction with adenovirus shLuc (control) and shGPS2. After 36 h, the cells were changed to Substrate-limited medium (Seahorse XF Base Medium plus 0.5 mM Glucose, 1.0 mM GlutaMAX, 0.5 mM Carnitine, and 1% FBS, pH 7.4) and incubated for overnight. The cells were pre-incubated in Fatty Acid Oxidation Assay Buffer (KHB Buffer (111 mM NaCl, 4.7 mM KCl, 1.25 mM $CaCl_2$, 2.0 mM $MgSO_4$, and 1.2 mM $NaH_2PO_4$) supplemented with 2.5 mM glucose, 0.5 mM carnitine, and 5 mM HEPES, pH 7.4) at 37 °C for 45 min in a non-$CO_2$ incubator. BSA (control) or BSA–palmitate were added just prior to starting the assay. After subsequent injecting and mixing of the compounds (assay concentrations: 2.0 µM oligomycin, 2.0 µM FCCP, and 0.5 µM Rotenone/antimycin A), the OCR was determined in all wells three times. The results were normalized with the protein quantity of each corresponding well.

**Metabolic studies in mice.** 7-8-week-old WT and LKO mice were fed with a 60% fat diet (HFD, Research Diets, D12492) or with a 10% fat diet (LFD, Research Diets, D12450) for 12 weeks. To establish a fibrosis model, mice were fed with a MCD (Research Diets, A02082002B) or with a control diet (Research Diets, A02082003B) for 4 weeks. During the experiments, body weight was assessed at different time points. OGTT and ITT were performed in the chow-diet and HFD feeding mice as previously described[21]. Corresponding blood glucose levels were measured at the indicated time points using a glucometer (Accu-Chek Performa, Roche). Plasma insulin (Mercodia, 10-1247-01), FGF21 (R&D Systems, DF2100), β-Hydroxybutyrate (Cayman, 700190) and hydroxyproline levels (Sigma, MAK008) were determined by ELISA.

Liver TG and cholesterol were extracted and measured with a colorimetric diagnostic kit (WAKO diagnostics) as previously described[21]. Plasma lipoproteins were fractionated from 2.5 µl of each sample using a Superose 6 PC 3.2/30 column (GE Healthcare) followed by online determination of TGs and cholesterols as previously described[21]. The lipid concentrations of the different lipoprotein fractions were calculated after integration of individual chromatograms.

Lipase activity is measured using the assay kit (Sigma Aldrich, MAK046) according to manufacturer instructions. In brief, 30 min after heparin injection, the mice were sacrificed and mouse plasma was separated by centrifuging for 10 min, 3000 r.p.m. at 4 °C. Plasma was used to test the lipase activity with (HL activity) or without 1 M NaCl, LPL was determined by substracting HL activity from the total lipase activity.

**Morphometric analyses.** The tissue processing and staining were performed in morphological phenotype analysis (FENO) core facility at Department of Laboratory Medicine, Karolinska Institute. Briefly, the mice liver tissue samples were fixed in 3% formaldehyde solution overnight and embedded in paraffin. Tissue slides were stained with H&E for the evaluation of the tissue morphology, with Sirius red for collagen-specific staining, or with F4/80 (Abcam, ab6649) for liver resident macrophages, following standardized protocols. Sirius red and F4/80 positive areas were quantified using imageJ.

All mice used in the studies were male, between 7 and 16 weeks old at the time of the experiment starting point, and randomized before any experiment was started. All animal experiments were approved by the respective national ethical boards (Swedish Board of Agriculture) and conducted in accordance with the guidelines stated in the International Guiding Principles for Biomedical Research Involving Animals, developed by the CIOMS. All mice strains were bred and maintained at the Center for Comparative Medicine at Karolinska Institutet and University Hospital (PKL, Huddinge, Sweden).

**Microarray analysis.** Raw-intensity expression files (.CEL files) (GSE83452; GSE48452; GSE73299; GSE49388; GSE54192) were imported to R and Bioconductor using the Oligo package[55]. The same package was used for quantile normalization, background correction, and summarization by robust multichip average preprocessing (RMA). The normalized log2-transformed expression values were then imported to the Limma[56] package for differential-expression analysis by linear modeling. A paired design was used to remove the batch effect between the biological replicates. Furthermore, genes with low expression (less than the 95th quantile of the negative-control probes) were removed. Genes with a P-value of <0.05, after adjusting for multiple hypothesis testing using the FDR method, were defined as being differentially expressed.

Correlation matrix was calculated based on expression values in log2 scale of 251 candidate NASH genes[15] and GPS2–NCOR–HDAC3 complex components from 104 NASH livers at baseline. Genes whose expression significantly correlated with GPS2 expression were selected and their correlation with GPS2–NCOR–HDAC3 complex components was plotted as heatmap.

**qPCR analysis.** Total RNA was extracted from snap-frozen liver tissues using Trizol Reagent (Thermo Fisher Scientific, 15596-026) and RNeasy RNA Mini Kit (Qiagen, 74104). Complementary DNAs were synthesized using M-MLV Reverse Transcriptase (Life Technologies, 28025-021). QPCR was performed using the ABI Prism 7500 PCR system (Applied Biosystems). 36b4 (gene encoding acidic ribosomal phosphoprotein P0, also called Rplp0) were used for normalization to quantify relative mRNA expression levels. Relative changes in mRNA expression were calculated using the comparative cycle method ($2^{-\Delta\Delta Ct}$). Primers are listed in Supplementary Table 1.

**RNA-seq.** RNA was extracted from mice liver biopsies as described above. RNA quality was assessed by 2200 TapeStation Instrument (Agilent). PolyA RNA selection was performed using the Illumina TruSeq RNA Sample Preparation Kit according to the manufacturer's protocol. RNA-seq libraries were prepared and sequenced on the Illumina HiSeq 2000 platform at Bioinformatics and Expression Analysis core facility (BEA, Karolinska Institutet, Sweden). Preprocessed reads were aligned to the mm9 transcriptome using the HISAT2 program, and Hypergeometric Optimization of Motif EnRichment (HOMER, http://homer.salk.edu/homer) was used to create the tag directory and count tags in all exons. Raw tag counts were imported in to R and Bioconductor and edgeR package was used to determine differential gene expression.

**ChIP and ChIP-seq sample preparation.** ChIP samples were prepared as described previously[21], with minor modifications. Briefly, fresh livers were chopped into small pieces and crosslinked with 1% formaldehyde (ThermoFisher, 28906) in PBS for 10 min for histone modifications, or double crosslinked with 2 mM disuccinimidyl glutarate (DSG) for 30 min, followed by 1% formaldehyde for 10 min, for TFs and GPS2. The reaction was stopped with glycine at a final concentration of 0.125 M for 5 min.

Liver pieces were disaggregated in ice-cold PBS with protease inhibitor using Dounce Homogenizer first with loose and later with tight pestle (Fisher Science, FB56691). Nuclei were isolated using lysis buffer 1 (50 mM Hepes–KOH, pH 7.5, 140 mM NaCl, 1 mM EDTA, 10% glycerol, 0.5% IGEPAL CA-630, and 0.25% Triton X-100), lysis buffer 2 (10 mM Tris–HCl, pH 8.0, 200 mM NaCl, 1 mM EDTA, and 0.5 mM EGTA), and lysis buffer 3 (10 mM Tris–HCl, pH 8.0, 100 mM NaCl, 1 mM EDTA, 0.5 mM EGTA, 0.1% Na-deoxycholate, and 0.5% N-Lauroylsarcosine), and subsequently sonicated for 30 min (30 s ON/30 s OFF) in the Bioruptor Pico (Diagenode). Protein A Dynabeads (Invitrogen) were incubated O/N with the antibodies. Each lysate was immunoprecipitated with the following antibodies: control rabbit IgG (Santa Cruz, sc-2027, 1–5 µg), anti-H3K4me3 (Abcam, ab8580, 1 µg), anti-H3K27ac (Abcam, ab4729, 1 µg), anti-PPARα (Millipore MAB3890, 5 µg), anti-Polymerase II (Biolegend, 664906, 5 µg), anti-NCOR (Bethyl laboratories, A301-145A, 4 µg), anti-SMRT (Bethyl laboratories, A301-147A, 4 µg), anti-HDAC3 (Santa Cruz, sc-11417, 5 µg), and anti-GPS2 (4 µg). Formaldehyde cross-linking was reversed overnight at 65 °C, and the immunoprecipitated DNA was purified using the QIAquick PCR purification kit (Qiagen). Primers for the ChIP qPCR are listed in Supplementary Table 1.

To prepare the ChIP-seq samples, the same ChIP protocol was followed, but using the ChIP DNA Clean and Concentrator Capped Zymo-Spin I (Zymo Research) purification kit. Two to four ChIPs were pooled during the final step of the purification to obtain concentrated material.

For library preparation and sequencing, 2–10 ng of ChIPed DNA was processed using Rubicon ThruPLEX DNA-seq kit (TAKARA) or processed at the EMBL Genomics Core Facility (Heidelberg, Germany) using standard protocols, and sequenced in the Illumina HiSeq 2000 (50SE reads, EMBL) or NextSeq 550 (75SE reads, BEA Core Facility, Karolinska Institutet, Sweden).

**ChIP-seq data analysis.** The computations were performed on resources provided by SNIC through Uppsala Multidisciplinary Center for Advanced Computational Science (UPPMAX) under Project SNIC 2018/8-122. Analysis was performed as previously described[21]. Sequencing files (fastq files), provided by the EMBL Genomics (Heidelberg, Germany) or BEA (Karolinska Institutet, Sweden) Core

Facility, and raw data from the published ChIP-seq data (PPARα: GSE61817; H3K4me1: GSE29184; NCOR: GSE26345; SMRT: GSE51045) were aligned to the NCBI37/mm9 version of the mouse reference genome, using Bowtie[57]. Peaks were identified using HOMER package[58].

Peak heights were normalized to the total number of uniquely mapped reads and displayed in integrative genomics viewer (IGV)[59] as the number of tags per 10 million tags. The sequences found in GPS2 peaks were subjected to motif analysis to identify potentially over-represented TF-binding sites using HOMER . Heat-map clustering of the peaks was performed in Cluster 3.0[60] using self-organizing maps, and then visualized in TreeView[61]. For statistical analysis of the peaks, raw tag counts were imported into R and Bioconductor and edgeR package was used to identify potential differential-binding sites[62,63].

**Western blot analysis**. Samples were lysed in RIPA buffer supplemented with protease and phosphatase inhibitors and were diluted to a concentration of 20 μg of protein and heated at 98 °C for 10 min. Proteins were separated by SDS–PAGE electrophoresis and transferred to PVDF membranes (Amersham International). Blocking reagent (SuperBlock™ T20 (PBS) blocking buffer, Thermo Fisher Scientific) were incubated for 1 h, and primary antibody was incubated overnight at 4 °C in the blocking solution. The antibodies and their concentrations are the following: anti-GPS2[38,43] (generated from Agrisera; 1:3000), anti-β-actin (Abcam, ab8226; 1: 30,000), anti-HDAC3 (Santa Cruz, sc11417; 1:3000), anti-PPARα (Cayman, 101710; 1:3000), anti-HA (BioLegend, 901514; 1:5000), and anti-Flag (Sigma-Aldrich, F7425; 1:5000). After several washes in PBST (PBS with 0.05% Tween 20), horseradish peroxidase (HRP)-labeled secondary antibodies (1:5000) were incubated for 1 h at room temperature in PBST. Membranes were developed with ECL western-blotting substrate (BioRad, 1705061).

**Co-immunoprecipitations**. HEK293 cells were co-transfected with (1) WT or truncated pcDNA3-HA-GPS2 and expression plasmids for Flag-tagged PPARα to mapping the interaction domain of GPS2 and PPARα or (2) WT or truncated HA-GPS2, HA-PPARα, and Flag-NCOR to determine the affinity of NCOR and PPARα in the absence and the presence of WT or truncated GPS2. Cells were lysed 48 h after transfection, and the lysate was incubated with anti-Flag (Sigma-Aldrich, F7425) coupled to protein A magnetic beads for 3 h, at 4 °C (15 μl beads were pre-incubated with 2 μg of antibody for 2 h, at 4 °C). Beads were washed with lysis buffer five times and eluted at 98 °C for 10 min. The eluted sample was loaded in an acrylamide gel by following the western blot protocol, and blotted with anti-Flag or anti-HA. Whole-cell lysis was used as input.

**Statistical analysis**. All the replicate experiments (including cell and mouse-based experiments) are biological replicates, which were repeated at least two times in the lab. Sample size was not pre-specified statistically. D'Agostino and Pearson normality test was used to determine the normal distribution. The variance within each group of data was compared using $F$-test (two groups) or Brown–Forsythe test (more than two groups). All statistical tests were performed using GraphPad Prism 6.0b (GraphPad Software, Inc., La Jolla, CA), and all data are represented as mean ± s.e.m. Statistical tests were assessed after confirming that the data met appropriate assumptions (normality, homogenous variance, and independent sampling). All statistical tests were two-tailed, and $p < 0.05$ was defined as significant. No statistical methods were used to predetermine sample size. No samples or animals were excluded from the analysis.

**Reporting summary**. Further information on experimental design is available in the Nature Research Reporting Summary linked to this article.

## Data availability
Gene expression RNA-seq data and ChIP-seq data have been deposited at the NCBI Gene Expression Omnibus (GEO) accession numbers are GSE113157. The authors declare that all data supporting the findings of this study are available in figshare with https://doi.org/10.6084/m9.figshare.7637504. Other data are available from the corresponding author upon request.

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

## Acknowledgements

We acknowledge Ozgene Pty, Ltd. for generating floxed GPS2 mice. We acknowledge V. Benes and the team at the EMBL Genomics Core Facility and F. Fagerström-Billai and co-workers at the BEA Core Facility (Karolinska Institutet) for ChIP-seq library preparation, sequencing, and analysis. We thank K. de Bosscher (Ghent University) and J. Taipale (Karolinska Institutet) for providing plasmids. E.T. was supported by grants from the Center for Innovative Medicine (CIMED) at the Karolinska Institutet, the Swedish Research Council, the Swedish Cancer Society, the Swedish Diabetes Foundation, the Novo Nordisk Foundation, and the European Union FP7 HEALTH project HUMAN. R. F. was supported by the European Foundation for the Study of Diabetes (EFSD)/Lilly research fellowship, Junior Diabetes Wellness Sverige grant and KI research foundation grants. R.K. and T.S. were supported by KI/SL and grants from CIMED. N.L. received a doctoral education grant (KID) from Karolinska Institutet. Z.H. was supported by Chinese Government Scholarship (CSC). N.V. was supported by grants from the French National Agency of Research (CONRAD and PROVIDE), Region Ile de France (CORDDIM), Paris city (EMERGENCE), the French Foundation for Diabetes (SFD), and the European Union H2020 framework (ERC-EpiFAT 725790). The human liver transcriptome study performed by P.L., F.A., N.V., and B.S. was supported by EU [HEPADIP (Contract LSHM-CT-2005-018734) and RESOLVE (Contract FP7-305707)], the European Research Council (ERC Grant Immunobile, contract 694717), Fondation pour la Recherche Médicale (Equipe labellisée, DEQ20150331724) and Agence Nationale pour la Recherche (ANR-10-LBEX-46). B.S. is a recipient of an Advanced ERC Grant (694717). We thank all members of our laboratories for scientific discussions and materials.

## Author contributions

R.F. and E.T. designed and supervised the study. N.L. and R.F. performed most experiments. A.D. and N.L. analyzed the NGS data. N.L., S.G., M.G., S.B., and R.F. performed the animal experiments. O.A., M.P., L.V., and P.P. analyzed the lipoprotein profiles and joined discussions. N.L., R.F., Z.H., and T.J. helped with the cloning and molecular experiments. A.M. and A.H. provided and processed the *Ncor* and *Smrt* KO livers. T.S. and R.K. analyzed the pathology. P.L., F.A., N.V., S.F., L.G., and B.S. provided and analyzed human data. R.F., N.L., and E.T. wrote the manuscript. All authors discussed the results and commented on the manuscript.

## Additional information

**Competing interests:** The authors declare no competing interests.

