## [Peer Review File · Nature Communications]

Reviewers' comments:

Reviewer #1 (Remarks to the Author):

The authors in the manuscript reported the phenotype of liver specific GPS2 KO mice. They found that liver KO of GPS2 improved MCD-induced fibrosis and obesity-related hepatic steatosis and insulin resistance. They also conducted gene expression studies and found that loss of GPS2 activated PPARalpha target gene. In addition, they showed that GPS2 bound to nuclear receptor elements. They also showed that the effect of GPS2 relied on PPARalpha. Finally, they provided evidence that GPS2 cooperated with NCoR but no SMRT in the liver.

The phenotype is novel and interesting. However, it is worth noting that although p-values may be significant, some of effects are modest. The main concern is the lack of in-depth mechanistic studies to explain how GPS2 inhibits PPARalpha target genes. Some of the effects described in the manuscript, including changes in epigenetic marks, may or may not be a direct consequence of the loss of GPS2. Although the authors showed an increase in PPARalpha binding, it is unclear how this led to activation of transcription because NCoR, SMRT or HDAC3 were not affected (Figure 6d). The authors should study other molecules to explain GPS2-dependent changes in gene expression. Finally, it is not clear how GPS2 works with NCoR rather than SMRT.

Specific points:

1. In Figure 1c, the correlation effects are not very convincing in the left two panels, although the p-value may be significant.
2. In Figure 4, the PPAR α expression level should be shown to exclude the possibility that its increased binding may be due to altered expression. Does the level of PPARalpha binding change globally ?
3. Figure 6d should also include PPARalpha.
4. The input signals should be included in all ChIP-Seq tracks.
5. Most studies reported correlative associations. The authors should conduct in vitro studies to independently confirm the results using cultured cells with siRNA or CRISPR/CAS9.

Reviewer #2 (Remarks to the Author):

Liang and colleagues have studied the influence of loss of the gene *Gps2* on the livers of mice and found that it improves features of NALFD via enhanced activity of PPAR α . A variety of data are presented covering both the metabolic impact and the molecular details of the interaction between PPAR α and *GPS2*. The results are novel and further clarify the complex transcription network involved in PPAR α -driven gene regulation. Overall, the manuscript is a bit messy and does not visualize the data in its optimal form, a problem that can be easily resolved by better structuring and attention to detail. My main concern relates to the relatively poor characterization of the NASH phenotype in the mice, which, given the use of NASH in the title, is an absolute must.

My suggestion is to change the title to do more justice to the actual data. A suggestion would be: "Loss of *Gps2* in mice reduces NASH via activation of PPAR α ." Or something similar.

The data in supplemental figure 1 do not strengthen the paper. The fact that a correlation meets statistical significance does not mean much. Please note that an r of 0.25 means an r -square of 0.0625. My suggestion would be to remove supplemental figure 1 or those correlations that barely meet statistical significance according to $P < 0.05$ (since multiple correlations were determined, multiple-testing correction should be applied anyway).

Please tone down the prelim conclusion on line 143. A more appropriate description would be: "GPS2 expression is correlated with expression of fibrotic markers in liver of patients with NASH, suggesting a potential role of GPS2 in NASH." Or something along those lines.

The observation that ketone body levels are elevated in *Gps2* LKO mice under fasting conditions suggests that loss of *Gps2* leads to enhanced PPAR α activity in the fasted state. It is important to know if hepatic expression of PPAR α target genes is higher in fasted *Gps2* LKO mice than in fasted WT mice.

Please be more specific with the description of the metabolic state of the animals. Mice depicted in supplemental figure 2g are fasted. For how long? What is the nutritional status of the mice shown in supplemental figure 2b-f?

How were the qPCR gene expression data normalized. It is a bit curious that the baseline expression values are about 1, but not exactly 1 (which would be expected). Please explain or correct in the figures.

The inflammatory phenotype of the Gps2 LKO is poorly explored. Please perform H&E stainings of the mice fed the MCD diet to determine the impact of Gps2 LKO on NASH phenotype. Are inflammatory infiltrates less visible in livers of Gps2 LKO mice? Also, qPCR analysis of inflammatory markers including macrophage markers such as Cd68 and F4/80 should be included. The rationale of the LPS treatment is not at all clear. My strong suggestion is to remove these data (supplemental figure 2l).

The data presented in supplemental figure 2 are from baseline mice for the LPS treatment. This does not make much sense. Instead, the expression of these genes should be measured in the liver samples of the mice shown in supplemental figure 2b-f.

The ITT data in supplemental figure 3m and figure 3e do not lead to the conclusion that insulin tolerance/sensitivity is better in the Gps2 LKO mice. The decrease in blood glucose in response to insulin, which reflects the sensitivity to insulin, is not changed between the two sets of mice. Please adjust.

The description on page 9 (line 188 to 204) is confusing because it goes back and forth between chow-fed and HFD-fed animals. Please be consistent. Start with chow fed and then move on to the mice fed HFD. Also, my suggestion is to move the data from supplemental figure 3 about mice fed HFD to regular figure 3.

Please be consistent with the legend for the WT and Gps2 LKO mice. In some figures (e.g. figure 3b), WT is white and KO is black. In other figures (e.g. figure 3d), it is the other way around.

Please comment on how liver specific Gps2 KO can influence adipose tissue mass and bodyweight gain. It is plausible that the metabolic improvements in Gps2 LKO mice are due to reduced adiposity.

The names of the genes shown in figure 4k are not readable. If the purpose was for the text to be readable, please adjust. If not, please remove the names.

Figure 4l is not convincing. Please remove. These data weaken the paper rather than strengthen it.

Line 764: WT should be PKO, LKO should PGKO.

The description of supplemental figure 4g is incorrect. Should be PPARA instead of PDK4.

The lack of increase in fasting induced ketone body production by loss of Gps2 in the absence of PPAR α is an important result that should be in the main figures. It is the key result that illustrates the physiological relevance of the purported interaction between PPAR α and GPS2.

Supplemental figure 5i are not publication quality. Please be gentle to the eyes of the reviewer.

Scientific writing does not allow for the use of “proof” and “prove”. Please use more scientific alternatives (evidence, demonstrate).

The Co-IP data showing a physical interaction between PPAR α and GPS2 should be moved to the main figures.

Reviewer #3 (Remarks to the Author):

Liang et al. submit a manuscript entitled “GPS2 accelerates the progression of 1 non-alcoholic steatohepatitis through PPAR α -selective mechanisms”. They first identify GPS2, a subunit of the NCoR/HDAC corepressor complex, in an analysis of published RNA-seq data from human liver biopsies. Even though GPS2 expression did not itself correlate significantly with NAFLD/NASH status, they noted that its expression nonetheless correlated with expression of fibrogenic genes. The authors then move to mouse models with liver-specific deletion (LKO) of GPS2 driven by albumin-Cre. When these mice were subject to MCD diet, the LKO mice were protected from liver steatosis, damage, and fibrosis, with gene expression suggesting higher fatty acid oxidation (FAO). When these mice were subjected to HFD, they were protected from body weight gain and, consistent with this, there was improvement in glucose tolerance and insulin sensitivity, as well as liver and serum TGs. RNA-seq of WT vs LKO mice showed dysregulation of multiple lipid metabolic pathways, and the authors focus on PPAR α as a regulator of FAO. Livers from LKO mice showed increased induction of PPAR α target genes upon fasting or treatment with the PPAR α agonist GW7647, yet no change in

gene activation by an LXR agonist. Furthermore, PPAR α occupancy by ChIP-qPCR at two strong binding sites (Pdk4 and Cyp4a14) was increased in LKO livers. In contrast to GPS2 LKO mice, double knockout mice lacking liver GPS2 and whole body PPAR α failed to show increased expression of PPAR α target genes, showing that GPS2 deficiency requires PPAR α to induce these genes. ChIP-seq for GPS2 was performed, showing enrichment of DR-1 PPAR motifs and occupancy at sites with enhancer and promoter histone marks, and changes in these marks upon GPS2 knockout that correlate with nearby gene expression. GPS2 recruitment to sites near Pdk4 and Cyp4a13 was reduced in livers lacking NCoR (NCOR1) but not SMRT (NCOR2), indicating an interesting selectivity of GPS2 for the NCOR1 corepressor complex. The overall mechanism proposed is that GPS2 knockout de-represses PPAR α to activate FAO and ketogenesis, thus decreasing liver steatosis and fibrosis.

This manuscript is overall well-written and very interesting. The statistical tests appear appropriate, as does the detail in the methods. The fact that GPS2 LKO protects against hepatic steatosis, while knockout of other subunits of the NCoR/HDAC3 complex cause steatosis, is alone a novel and exciting result. The authors present some very good evidence supporting the mechanism of de-repression of PPAR α target genes, but additional key experiments (see below) would further bolster the model.

General comments:

1) The first part of the paper in human liver samples nicely shows variable expression of GPS2. Are the genetic variants (i.e. eQTLs) associated with GPS expression? The authors should interrogate the GTEx dataset to address this. The other possibility is that GPS2 expression changes in the setting of liver steatosis. For this reason, the authors should report whether mouse liver GPS2 expression changes in their MCD and HFD models.

2) The GPS2 LKO mouse model is key to this manuscript, as is the successful ChIP for GPS2 in mouse liver. The tissue-specific knockout of GPS2 is shown nicely at the mRNA level and protein levels (Fig 2B-C), and should also be shown at the level of the ChIP assay (i.e. GPS2 ChIP occupancy is decreased/absent in the LKO). Furthermore, the authors state that other components of the NCOR/HDAC3 complex do not change upon GPS2 LKO based on mRNA levels in Supplementary Fig 2B, but to make this conclusion Western blots are necessary to show unchanged protein levels.

3) For many of the bar graphs (i.e. liver TG in Fig 2G and 3H, etc), it would be preferable to show the individual data points for each mouse as well as the mean/sem. I'm not sure whether Nature Communications has this requirement, but it's generally considered good practice today.

4) The serum ketone data (Supp Fig 2G and 5E) is impressive and highly relevant, as PPAR α is the key regulator of ketogenesis. I recommend moving this data to the main rather than supplemental figures. (Also adding Hmgcs2 to panel of mRNAs measured would bolster this.) The changes in ketones are the only functional readout for the proposed mechanism: that GPS2 knockout de-represses PPAR α to activate FAO and ketogenesis, thus decreasing liver fat and VLDL secretion. Ideally, one other other functional assay supporting this model should be included (i.e. direct measurement of FAO, indirect measurement like acyl-carnitines, measurement of VLDL secretion, etc).

5) The authors propose that GPS2 in liver lacks the anti-inflammatory properties it has in other tissues, but this was only shown after the major inflammatory stimulus of LPS treatment. Liver inflammation assays should also be reported in their MCD and HFD models of more “metabolic inflammation.”

Major issues:

1) In the MCD experiments in Figures 2D-H, a control diet group is missing. For all the improvements seen in GPS2 LKO mice (lower transaminases, altered gene expression, liver TGs, Sirius red), it's unclear how much improvement occurred and whether it approaches normal liver. If this data exists, I recommend inclusion (the gene expression differences on a control diet are indeed in supp Fig 2M). Another issue is the panel of genes in Figure 2F. Were other fibrotic and lipid metabolic genes tested and showed no difference? These could be shown in supplemental. Furthermore, this panels lacks any inflammatory genes, and inflammation is a key aspect of NASH that is induced in the MCD model. Therefore, it would be of great value to include inflammatory genes like ILs, TNF, etc in this panel (see comment 5 above). A key point in Supp Fig 2I is that these genes did not differ upon LPS treatment (so GPS2 has a selective role in fibrosis), but it is critical to know in the MCD model whether GPS2 LKO is anti-inflammatory. Finally, for the key point of reduced fibrosis, some quantification is necessary beyond the gene expression in Fig 2F and the staining in Fig 2H. The Sirius red staining (% of area) is routinely quantified, and hydroxyproline is a standard biochemical assay of collagen deposition.

2) In the text describing the HFD diet model in Figure 3, it must be more explicitly acknowledged that that the decreased body weight and adiposity of the LKO mice is alone consistent with all the metabolic improvements observed. Therefore, the concluding sentence (line 214-215) that hepatocyte GPS2 modulates lipid/glucose homeostasis needs to be qualified by stating that weight is reduced. It is an interesting question but beyond the scope of this study to determine why less weight is gained (i.e. metabolic cage analysis of energy expenditure, etc). Finally, given that liver fibrosis is described as the key role of GPS2, no assays of fibrosis (gene expression, staining) were

reported on this HFD model. This is likely because not much fibrosis is induced by this diet, but the point should at least be mentioned (“In contrast to the MCD, no difference in fibrotic signature was observed in GPS2 LKO in this HFD model, which failed to trigger much fibrosis.”)

3) The PPARα CHIP data in Fig 4H-I is missing the key negative control. While IgG is okay, it would be preferable to show the PPARα CHIP signal is at a background genomic site without any binding. Especially given the low percentage of input at such strong sites (Pdk4 and Cyp4a14), Q-PCR primers for a negative control site are essential here. Another potential negative control for the PPARα CHIP is using the PPARα knockout livers described later. Likewise, other CHIP-QPCR experiments in Figure 5-6 would also benefit from the negative control of a non-binding site, to see the degree of enrichment of binding sites over background in the CHIP DNA.

4) Figure 5 is confusing and mostly unnecessary. The figure title says that “GPS2 deletion causes epigenetic activation”, but the data show apparently similar amounts of “inactivation” (i.e. genes and histone marks that go down with GPS2 deletion). The correlations in Figures 5C-H simply show that changes (up or down) in activating histone marks and nearby gene expression generally correlate with one another (globally and at individual loci), which is always seen and reveals nothing about GPS2 biology. Such data could be supplemental. The most potentially relevant result with PPARα knockout is in Fig 5I, but has the same issue as above missing the negative control site without H3K27ac, and even more problematic it’s missing the key groups with PPARα present (and presumably higher H3K27ac at these strong target genes). To correctly interpret this experiment two more groups are needed: wild type and GPS2 deletion only (comparing these to show that there’s a statistically significant increase in acetylation detectable by this CHIP QPCR assay). Without this, it’s difficult to make the conclusion that GPS2 needs PPARα to affect the epigenome. Rather than just looking at the epigenomic markers, an even better test of the authors’ model would be looking at GPS2 genomic occupancy in their same PPARα knockout mice – ideally by CHIP-seq but at least by CHIP-QPCR at selected sites (analogous to the GPS2 CHIP in the NCOR1/2 knockout mice in Fig 6E). Showing a loss of GPS2 occupancy in the absence of PPARα is essential to the overall model.

Minor issues:

line 2: In the title, I recommend “PPARα-dependent” rather than “-selective”. A lot of data in double knockouts supports the necessity of PPARα, but only the supplemental experiment with an LXR agonist (Suppl Fig 4H) supports selectivity, and other nuclear receptors and transcription factors are not addressed.

line 28: in the abstract, the authors should clearly identify GPS as “a subunit of the NCOR/HDAC3 nuclear receptor corepressor complex”, or something to that effect.

line 74: rather than “a fundamental co-repressor complex”, the name “NCOR/HDAC3 corepressor complex” should be used instead (like it is in the discussion), with mention that it’s the major corepressor for nuclear receptors, and also for other TFs.

lines 76-80: I would identify NCOR1/2 as the main structural components of the complex, HDAC3 as the catalytic one, and TBLs and GPS2 as other small subunits of uncertain function – then this manuscript does a nice job ascribing some function to GPS2.

line 122: This sentence identifies a human NASH signature gene set. Was this gene list published in the original papers? Or did it emerge from this re-analysis? If the latter, then it would be useful to include the gene lists in a supplementary table.

line 123: The critical result that “expression of GPS2 was not different in NASH liver” is stated but no data is shown. The (negative) data supporting this important point should at least be in the supplemental material.

line 124: Even though the list of GPS-correlated genes is on the x-axis of Fig 1B, it would also be useful to put these in a supplemental table.

line 139: In Figure 1G, it is unclear whether the x-axis refers to GPS2 mRNA levels before or after weight loss. The same issue applies to the analyses in Suppl Figures 1F-I.

line 149: The call-out for tissue-specific knockout say Supp Fig 2B, but it should refer to main Fig 2B.

line 150: There is no description of data in Supp Fig 2B, so the end of this sentence should include a clause like “without effect on mRNA levels for other subunits of the NCoR/HDAC3 complex (Supp Fig 2B).”

line 152: Rather than “lipoprotein profiles”, just state that TG in the VLDL fraction and the total TG were decreased.

line 158: conclude the sentence about unchanged lipases with the explanation: “suggesting decreased hepatic VLDL production rather than increased lipase activity.”

line 220: For this RNA-seq, I presume the mice were on control/chow diets but it’s not apparent from the text, figure legend, or methods. Also, in describing other transcriptome data the authors nicely give the number of genes up- and down-regulated. It looks like a lot based on the heatmap in Fig 4A, but numbers and a gene list in the supplementary data are advised.

line 239: the text says “enhanced response to...agonist” yet the data in Figure 4D is only the presence of agonist. It would be better for this graph to include the untreated group to see the induction by agonist in each genotype. Also, the y-axis is “relative gene expression” which indicates that one condition has been set equal to 1 for each gene (i.e. in Fig 4D it appears that WT fed is the normalizer). Especially from the Pdk4 data it’s clear that there’s no normalizer condition on the current graph in 4D.

line 244: The PPARa CHIP-seq data in Fig 4F-G appear at first as it was done by the authors, but the methods make clear it was published by others, therefore the citation to Lee et al, Nature 516:112 should be included. Furthermore, browser tracks at two loci are used to make the case that GPS2 and PPARa co-occupy sites in liver. A global analysis of the CHIP-seq peaks (i.e. Venn diagram overlap, scatterplot, etc) would be better to make this point (i.e. “85% of PPARa peaks identified by others show GPS binding signal in our data”).

Line 255: for clarity in this section, “LKO” should be qualified as “GPS2 LKO” to avoid confusion with the PPARa whole body knockout.

line 256: wrong punctuation in middle of line, comma not period.

line 268: PDK4 is the only gene shown to make the case that GPS2 express negatively correlates with PPARA target gene expression in human liver. There are many other classic PPARA target genes measured on these arrays that could be correlated to make this this case. It would be preferable to see a heatmap-type analysis showing correlation of multiple PPARA targets with GPS2 expression.

line 417: The word “in” should not be italicized.

In Fig 2G, is the y-axis correct? 800ug/mg implies that the liver is 80% fat by weight? Another reason to include control diet livers in this analysis as above.

In supplementary figure 5, there are two "E" panels and no "D".

In supplementary figure 4, panels G and H should be later to preserve the order they are described in the text.

General response to the reviewers comments

We thank all three reviewers for their very supportive comments and appreciate the intriguing issues raised and detailed suggestions made. We have carefully addressed each of these in our extensive experimental revision as outlined in the point-by-point response. We believe that with the inclusion of the revised experimental data we could further improve the quality, the clarity and the conceptual impact of our study.

We would like to highlight the following key improvements of issues raised by all reviewers:

We have further analyzed the inflammatory phenotype of the HFD and MCD mice by integrating the H&E staining and F4/80 staining along with pro-inflammatory gene expression in both WT and LKO mice. This analysis revealed that ablation of GPS2 led to lipid metabolic changes linked with reduced macrophage infiltration and inflammatory gene expression in the liver, which might contribute to alleviated liver fibrosis.

We have performed additional experiments that together with the initially submitted data should further increase the mechanistic insights into GPS2 action. We have in particular generated new high-quality ChIP-seq data of PPAR α , GPS2 and NCOR from the livers of WT mice and all three KO models. These new data provide a unique resource and to our knowledge the first reported ChIP-seq analysis of corepressor recruitment in PPAR α KO mice. Also, the antibody/ChIP-seq signal specificity for each protein was confirmed by the lack of signals in the respective KO mice.

The following results are key to possible mechanisms of GPS2 repression: (1) Repression of PPAR α -target genes in liver involves physical interactions of GPS2 with PPAR α . (2) Repression of PPAR α -target genes in liver involves physical interactions of GPS2 with NCOR (and thereby likely with the HDAC3 corepressor complex, while SMRT is not required). (3) The GPS2-NCOR partnership is confirmed by increased H3K27ac at PPAR α target loci, suggesting epigenomic activation to be linked to de-repression. (4) Co-immunoprecipitations using WT and interaction-deficient GPS2 variants provide new evidence for ternary interactions between PPAR α , GPS2, and NCOR. (5) Consistent with the model that GPS2 and NCOR cooperate in PPAR α repression, our new PPAR α ChIP-seq data in *Gps2* and *Ncor* LKO livers further support the initial ChIP qPCR data that removal of either NCOR or GPS2 causes an increase of PPAR α recruitment at the *Pdk4* locus and genome-wide. (6) We have performed Pol II ChIP-seq in *Gps2* WT and LKO mice that demonstrates increased (presumably stalled) Pol II enrichment at the *Pdk4* and *Cyp4a14* transcription start site.

Overall, we believe that our work provides better mechanistic explanations for the co-regulation of PPAR α pathways by GPS2 and NCOR in hepatocytes. The proposal of functional sub-complexes, i.e. GPS2/NCOR (metabolic) vs. GPS2/SMRT (anti-inflammatory), conceptually advances our understanding of how 'the corepressor complex' functions to regulate tissue-specific metabolism, and more generally represses transcription.

Itemized list of all changes made in the revised manuscript

- We have re-organized the revised manuscript by moving the human data analysis to the end: old Figure 1 and Supplementary Figure 1 are now revised Figure 6 and Supplementary Figure 6). Therefore, all revised Figure numbers changed accordingly.
- **Revised Figure 1b (Old Sup Fig.2g), 1d (Old Fig.2e), 1e (Old Fig.2e), 1g (Old Fig.2g), 2b(Old Fig.3b), 2c(Old Fig.3c), 2f (Old Fig.3g), 2h (Old Fig.3h); Supplementary Figure 1e (Old Sup Fig.2d), 1g (Old Sup Fig.2h), 1h (Old Sup Fig.2i), 1i (Old Fig.2d), 1k (Old Sup.2j), 1l (Old Sup Fig.2k), 2a-e (Old Sup Fig 3.e-i), 2m-o (Old Sup Fig.3d, 3n, 3o).** We have now included the CD groups and revised the bar plots into dot plots.
- **Revised Figure 1m (Old Fig. 2f) and 2k (Old Fig.3f).** We have also included the panel of inflammatory genes in the qPCR analysis to demonstrate reduced inflammation in the HFD and MCD mice.
- **Added Figure 1c.** We have now included the Seahorse respiratory analysis in WT and LKO livers to demonstrate that GPS2 depletion enhances oxygen respiratory rate, directly supporting the *in vivo* observations of increased ketone bodies in LKO mice.
- **Added Figure 1f, 1h, 1i, 1j, 1k, 1l, 2g, 2h, 2i and 2j.** We have now included the H&E staining of the mouse livers, the quantification of the sirius red staining and hydroxyproline test data to further elucidate reduced fibrosis in LKO mice fed with MCD or HFD. We have also included the F4/80 staining and quantification.
- **Added Supplementary Figure 1j, 2j.** We have now included qPCR data to analyze mRNA expression changes of the corepressor complex subunits upon MCD and HFD feeding.
- **Added Supplementary Figure 2p.** We have added the qPCR analysis of fibrotic genes upon HFD feeding.
- **Revised Figure 3c, 3d (Old Fig. 4c and 4d).** We now added the control groups (fed, vehicle) in the qPCR analysis of target genes.
- **Added Figure 4f.** We have generated new ChIP-seq data and compared GPS2 and PPAR α peaks in mouse livers and find that 75% of PPAR α peaks are co-localized with GPS2.
- **Added Supplementary Figure 5j and 5k.** We have performed PPAR α ChIP-seq in WT, *Gps2* and *Ncor* LKO livers and find increased recruitment of PPAR α to GPS2- repressed gene loci upon depletion of either GPS2 or NCOR.
- **Revised Figure 4g (Old Fig.5i).** We have now included a H3K27ac ChIP-qPCR analysis at the *Pdk4* and *Cyp4a14* loci in WT, *Gps2* and *PPAR α* single and double KO mice.
- **Revised Supplementary Figure 4m, 4n (Old Sup Fig.5i).** We have now increased the resolution of the figures to make them readable.

- **Added Figure 5a, 5b.** We have performed H3K27ac ChIP-seq in WT, *Gps2* LKO, *Ncor* LKO and *Smrt* LKO livers. We observed a similar H3K27ac increase at *Pdk4* and *Cyp4a14* loci in *Gps2* and *Ncor* LKO livers but not *Smrt* LKO livers.
- **Added Figure 5c, 5d and Supplementary Figure 5g.** We have performed new ChIP-seq of PPAR α , GPS2 and NCOR in livers from WT mice and from all three respective KO mice to elucidate their interplay in the genomic level.
- **Added Figure 5e.** We performed co-immunoprecipitation of PPAR α and NCOR in the presence of full-length or interaction domain-truncated GPS2 variants. We found that full-length GPS2 enhances interactions of PPAR α with NCOR, suggesting ternary interactions of all three proteins.
- **Added Figure 5f.** We have performed Pol II ChIP-seq in *Gps2* WT and LKO mice to show that GPS2 ablation increased Pol II binding in *Pdk4* and *Cyp4a14* loci.
- **Revised Figure 6h (Old Fig.4k).** We now remove the gene list in the correlation panel as suggested by the reviewers.
- **Added Supplementary Figure 6a.** We added the GPS2 expression in different NASH stages as suggested by the reviewers.

Point-by-point response to the reviewer's comments

Our responses are indicated in RED.

Reviewers' comments:

Reviewer #1 (Remarks to the Author):

The authors in the manuscript reported the phenotype of liver specific GPS2 KO mice. They found that liver KO of GPS2 improved MCD-induced fibrosis and obesity-related hepatic steatosis and insulin resistance. They also conducted gene expression studies and found that loss of GPS2 activated PPARAlpha target gene. In addition, they showed that GPS2 bound to nuclear receptor elements. They also showed that the effect of GPS2 relied on PPARAlpha. Finally, they provided evidence that GPS2 cooperated with NCoR but no SMRT in the liver. The phenotype is novel and interesting.

However, it is worth noting that although p-values may be significant, some of effects are modest.

Reply: We agree with the concern and have addressed this in the revised Figure 6 (see our response to Specific Point 1 below).

The main concern is the lack of in-depth mechanistic studies to explain how GPS2 inhibits PPARAlpha target genes. Some of the effects described in the manuscript, including changes in epigenetic marks, may or may not be a direct consequence of the loss of GPS2. Although the authors showed an increase in PPARAlpha binding, it is unclear how this led to activation of transcription because NCoR, SMRT or HDAC3 were not affected (Figure 6d). The authors should study other molecules to explain GPS2-dependent changes in gene expression. Finally, it is not clear how GPS2 works with NCoR rather than SMRT.

Reply: We have performed additional experiments that together with the initially submitted data should further increase the mechanistic insights into GPS2 action. We have in particular generated new high-quality ChIP-seq data of PPAR α , GPS2 and NCOR from the livers of WT mice and all three KO model and performed new co-immunoprecipitations to elucidate the interplay of the three proteins, and we have analyzed Pol II recruitment by Chip-seq in WT and *Gps2* LKO mice (Figure 5 and Supplementary Figure 5).

The following results are key to possible mechanisms of GPS2 repression:

First, repression of PPAR α -target genes in liver involves physical interactions of GPS2 with PPAR α . We would like to emphasize this point as it is commonly not know that GPS2 directly binds to nuclear receptors, in addition to NCOR and SMRT. This is supported by co-immunoprecipitations (Figure 3f) where we also mapped the GPS2 domain (aa 105-155) that is crucial for the PPAR α interaction. In fact, GPS2 (aa 105-327) was isolated (but not published) in the initial yeast two-hybrid screens for PPAR α -interacting proteins from a human liver library (described in Treuter *et al. Molecular Endocrinology 1998*). Interaction is further supported by comparison of GPS2 and PPAR α ChIP-seq (cistrome) data which showed around 75% overlap (Figure 4f). Furthermore, the new GPS2 ChIP-seq data in WT versus PPAR α KO livers (Figure 5c and 5d, Supplementary Figure 5g) show that GPS

recruitment to promoters and enhancers of the *Pdk4*, *Cyp4a14* and *Acot* loci was strongly reduced in the absence of PPAR α .

Second, repression of PPAR α -target genes in liver involves physical interactions of GPS2 with NCOR (and thereby likely with the HDAC3 corepressor complex). This is clearly supported by the new ChIP-seq data along with the ChIP qPCR analysis at single promoters/enhancers (**Figure 5 and Supplementary Figure 5**), demonstrating that GPS2 recruitment is lost in NCOR KO livers.

Third, we provide new evidence for ternary interactions between PPAR α , GPS2, and NCOR in **Figure 5e**. We performed co-immunoprecipitation of PPAR α and NCOR in the presence of full-length or truncated GPS2 variants that lack the interaction domain with PPAR α or with NCOR, respectively. We found that full-length GPS2, but not truncated variants, enhances interactions of PPAR α with NCOR, which could contribute to the de-repression mechanisms upon GPS2 depletion.

Fourth, consistent with the model that GPS2 and NCOR cooperate in PPAR α repression, our new PPAR α ChIP-seq data in *Gps2* and *Ncor* LKO livers further support the initial ChIP qPCR data that removal of either NCOR or GPS2 causes a modest increase of PPAR α recruitment at the *Pdk4* locus (**Figure 5 c**) and genome-wide (**Supplementary Fig. 5 j, k**). Perhaps, lack of corepressor interactions (in the LKO livers) would stabilize PPAR α -chromatin interactions, comparable to ligand activation of PPAR α that also leads to corepressor release and increased chromatin binding.

Fifth, have performed Pol II ChIP-seq in *Gps2* WT and LKO mice that demonstrates increased Pol II enrichment (the peak indicates presumably a stalled Pol II) at the *Pdk4* and *Cyp4a14* transcription start site (**Figure 5f**). This suggest that de-repression (by GPS2 depletion) and further activation of PPAR α (by fasting or ligand treatment) may occur partially at the level of transcriptional elongation and be part of the corepressor-coactivator switching mechanism.

The authors should study other molecules to explain GPS2-dependent changes in gene expression.

Reply: We believe that the above-discussed revised data provide sufficient mechanistic details that do not necessarily require the postulation of other molecules (proteins). We do, however, not exclude that additional cofactors or PTMs may further specify GPS2 action in the liver. To identify candidates, IP-MS approaches as those recently described for liver HDAC3 (*Armour SM et al Nat Commun 8, 2017*) would be needed, but we have not fully established this for GPS2 and cannot achieve it within this revision.

Finally, it is not clear how GPS2 works with NCoR rather than SMRT.

Reply: The cooperation of NCOR but not SMRT with GPS2 is strongly supported by above described ChIP-seq and ChIP-qPCR data (**Figure 5 and Supplementary Figure 5**). We included new H3K27ac ChIP-seq data from WT, *Gps2* LKO, *Ncor* LKO and *Smrt* LKO livers further supporting that depletion of GPS2 and NCOR results in a similar increase of H3K27ac at *Pdk4* and *Cyp4a14* loci, while depletion of SMRT has no effect (**Figure 5 a,b**). We also provide evidence that NCOR and SMRT regulate very distinctive pathways in the liver since

the transcriptome signatures in *Ncor* and *Smrt* LKO livers are very different (**Supplementary Figure 5 c-f**).

Despite the new data the molecular reasons why SMRT is not involved in the liver PPAR α -GPS2 pathway remains enigmatic. GPS2 interacts with both NCOR and SMRT *in vitro* and co-localizes at liver chromatin. However, our findings are intriguingly consistent with previous data from Lazar and coworkers (Sun Z et al. Mol Cell 2013). They have shown that liver-specific knockout of NCOR, but not SMRT, causes metabolic and transcriptomal alterations resembling those of mice without hepatic HDAC3. They further suggest that interaction with NCOR, but not SMRT, is essential for deacetylase-independent function of HDAC3 in liver. As with GPS2, these data cannot be explained by the *in vitro* interactions of the ChIP-seq profiles of SMRT vs. NCOR.

Therefore, we believe that there is an enigmatic cell-type selective ‘corepressor sub-complex’ code which cannot be explained with the currently available data. Possible factors influencing the partnership of the subunits are cell-type selective TF profiles and post-translational modifications that modulate TF-corepressor interactions (e.g. PPAR-NCOR vs. PPAR-SMRT) and subunit interactions (GPS2-NCOR vs. GPS2-SMRT) *in vivo* at chromatin. SMRT and NCOR are sufficiently different to have different TF affinities. In fact, we have shown that in macrophages GPS2 works with SMRT but not with NCOR to repress inflammatory gene expression, in part via cooperating with c-jun but not with nuclear receptors (*Fan et al. Nature Med 2016, Treuter et al., FEBS letters 2017*). Thus, combining the liver and macrophage data about possible corepressor ‘sub-complexes’, it is possible that NCOR has a greater *in vivo* preference for nuclear receptors than SMRT. A further experimental exploitation of the respective conditional KO models would be required to test this, along with the determination of post-translational modifications and IP-proteomics at chromatin.

Specific points:

1. In Figure 1c, the correlation effects are not very convincing in the left two panels, although the p-value may be significant.

Reply: We have now removed the weak correlations ($\rho < 0.3$, related to old Supplementary Figures 1a,b,c,e,f,g,h,i) from revised **Figure 6** and **Supplementary Figure 6**. The old Figure 1c right panels are now moved to the revised **Supplementary Figure 6**.

2. In Figure 4, the PPAR α expression level should be shown to exclude the possibility that its increased binding may be due to altered expression. Does the level of PPARalpha binding change globally ?

Reply: We show in the revised manuscript in **Figure 3 c-d** that PPAR α expression does not change between WT and GPS2 KO livers. We also performed western blot (**Supplementary Figure 1b**) to show that PPAR α protein levels are not different between WT and LKO mice. The global PPAR α ChIP-seq analysis in the revised manuscript (**Supplementary Figure 5k**) shows that PPAR α binding is significantly higher at GPS2-repressed gene loci compared to loci of unchanged genes.

3. Figure 6d should also include PPARalpha.

Reply: We performed PPAR α ChIP-seq and included the new data in **Figure 5c-d**.

4. The input signals should be included in all ChIP-Seq tracks.

Reply: We included now two types of controls in most ChIP-seq data of the revised manuscript: 1) The input signals and 2) GPS2 ChIP-seq in WT and *Gps2* LKO livers and PPAR α ChIP-seq in WT and PPAR α KO livers (**Figure 4b-c, 5a-d**).

5. Most studies reported correlative associations. The authors should conduct *in vitro* studies to independently confirm the results using cultured cells with siRNA or CRISPR/CAS9.

Reply: We agree that the correlation associations from the clinical samples (**old Figure 1, revised Figure 6**) should optimally be supported by experiments in human hepatocyte/cell line models *in vitro*. However, it is currently difficult to recapitulate liver fibrosis upon GPS2 KO using an *in vitro* model. In addition, the TGF β and TIMP1 correlation with GPS2 only appears *in vivo*, upon challenge with fibrogenic diet, and depletion of GPS2 alone does not induce TGF β and TIMP1 in the chow diet mice, suggesting that GPS2 regulation of TGF β and TIMP1 is not a direct effect but rather secondary to affected liver lipid metabolism. The activation of Kupffer cells and stellate cells by altered liver lipid metabolism can also not be mimicked using cell lines. However, we performed *in vitro* fatty acid oxidation assay using Seahorse (**Figure 1c**) and indeed see an elevated fatty acid oxidation in the AML cell lines, which supports our hypothesis and is consistent with the phenotypes observed in the mice.

Reviewer #2 (Remarks to the Author):

Liang and colleagues have studied the influence of loss of the gene *Gps2* on the livers of mice and found that it improves features of NALFD via enhanced activity of PPAR α . A variety of data are presented covering both the metabolic impact and the molecular details of the interaction between PPAR α and GPS2. **The results are novel and further clarify the complex transcription network involved in PPAR α -driven gene regulation. Overall, the manuscript is a bit messy and does not visualize the data in its optimal form, a problem that can be easily resolved by better structuring and attention to detail. My main concern relates to the relatively poor characterization of the NASH phenotype in the mice, which, given the use of NASH in the title, is an absolute must.**

My suggestion is to change the title to do more justice to the actual data. A suggestion would be: "Loss of *Gps2* in mice reduces NASH via activation of PPAR α ." Or something similar.

Reply: We appreciate the suggestion and have changed the title to 'Hepatocyte-specific loss of GPS2 in mice reduces non-alcoholic steatohepatitis via activation of PPAR α '.

The data in supplemental figure 1 do not strengthen the paper. The fact that a correlation meets statistical significance does not mean much. Please note that an r of 0.25 means an r^2 of 0.0625. My suggestion would be to remove supplemental figure 1 or those correlations that barely meet statistical significance according to $P < 0.05$ (since multiple correlations were determined, multiple-testing correction should be applied anyway).

Reply: We have now removed the weak correlations ($\rho < 0.3$, related to old Supplementary Figures 1a,b,c,e,f,g,h,i) from revised **Figure 6** and **Supplementary Figure 6**.

Please tone down the prelim conclusion on line 143. A more appropriate description would be: "GPS2 expression is correlated with expression of fibrotic markers in liver of patients with NASH, suggesting a potential role of GPS2 in NASH." Or something along those lines.

Reply: We have revised the text accordingly.

The observation that ketone body levels are elevated in *Gps2* LKO mice under fasting conditions suggests that loss of *Gps2* leads to enhanced PPAR α activity in the fasted state. It is important to know if hepatic expression of PPAR α target genes is higher in fasted *Gps2* LKO mice than in fasted WT mice.

Reply: We include these data in the revised manuscript in Figure 3c-d.

Please be more specific with the description of the metabolic state of the animals. Mice depicted in supplemental figure 2g are fasted. For how long? What is the nutritional status of the mice shown in supplemental figure 2b-f?

Reply: We have revised the Figure legends accordingly in Supplementary Figure 1, 2 and 4.

How were the qPCR gene expression data normalized. It is a bit curious that the baseline expression values are about 1, but not exactly 1 (which would be expected). Please explain or correct in the figures.

Reply: The baseline expression is all normalized to the first mouse in the control group. The expression values are not exactly 1 because of the variations between individual mice in the same group.

The inflammatory phenotype of the *Gps2* LKO is poorly explored. Please perform H&E stainings of the mice fed the MCD diet to determine the impact of *Gps2* LKO on NASH phenotype. Are inflammatory infiltrates less visible in livers of *Gps2* LKO mice? Also, qPCR analysis of inflammatory markers including macrophage markers such as Cd68 and F4/80 should be included. The rationale of the LPS treatment is not at all clear. My strong suggestion is to remove these data (supplemental figure 2l).

The data presented in supplemental figure 2 are from baseline mice for the LPS treatment. This does not make much sense. Instead, the expression of these genes should be measured in the liver samples of the mice shown in supplemental figure 2b-f.

Reply: We include now the H&E staining for both MCD and HFD mice in Figure 1f and Figure 2g. We also include the F4/80 staining of both MCD and HFD mice in Figure 1k, 1l and Figure 2i, 2j. The analysis of inflammatory gene expression (including F4/80) is included in the revised manuscript of Figure 1m and Figure 2k.

We have included the LPS data as we had previously identified that GPS2 KO in macrophages increased inflammatory responses (Fan et al. Nat Med 2016). Since liver inflammation is correlated with NAFLD, we wanted to elucidate whether hepatocyte GPS2 KO promotes inflammatory responses in the liver. The results suggest that this is not the case and should be therefore still of interest.

For the basal fibrosis markers, we wanted to check if the liver fibrosis markers (such as TGF β and TIMP1) that are correlated with GPS2 expression in the human population are direct targets of GPS2. The unchanged fibrosis markers in the baseline suggested that they are most likely secondary effects to the dysregulated lipid metabolism in liver. We included the fibrosis marker qPCR analysis in the HFD mice in Supplementary Figure 2p.

The ITT data in supplemental figure 3m and figure 3e do not lead to the conclusion that insulin tolerance/sensitivity is better in the *Gps2* LKO mice. The decrease in blood glucose in response to insulin, which reflects the sensitivity to insulin, is not changed between the two sets of mice. Please adjust.

Reply: We have changed ‘improved glucose intolerance and insulin sensitivity’ into ‘improved glucose control’ in the revised manuscript.

The description on page 9 (line 188 to 204) is confusing because it goes back and forth between chow-fed and HFD-fed animals. Please be consistent. Start with chow fed and then move on to the mice fed HFD. Also, my suggestion is to move the data from supplemental figure 3 about mice fed HFD to regular figure 3.

Reply: The description is now adjusted accordingly.

Please be consistent with the legend for the WT and Gps2 LKO mice. In some figures (e.g. figure 3b), WT is white and KO is black. In other figures (e.g. figure 3d), it is the other way around.

Reply: The bar figures are changed accordingly.

Please comment on how liver specific Gps2 KO can influence adipose tissue mass and bodyweight gain. It is plausible that the metabolic improvements in Gps2 LKO mice are due to reduced adiposity.

Reply: We hypothesize that the reduced adipose tissue mass and body weight gain is largely due to enhanced *Fgf21* expression in *Gps2* LKO mice. As FGF21 promotes adipose tissue browning and promotes fatty acid catabolism. Indeed, basal FGF21 expression in the LKO mice almost reaches the fasting or GW (PPAR α agonist) levels (Figure 3c and 3d**). FGF21 is a direct target of GPS2 which is dependent on PPAR α (**Figure 3g, Supplement Figure 3b, 3c**). Indeed, LKO mice in the HFD mice showed increased expression of adipose tissue (VAT and SAT) browning markers (see the attached file).**

The names of the genes shown in figure 4k are not readable. If the purpose was for the text to be readable, please adjust. If not, please remove the names.

Reply: We removed the gene names in the revised manuscript (Figure 6h).

Figure 4l is not convincing. Please remove. These data weaken the paper rather than strengthen it.

Reply: We removed the data from the revised manuscript.

Line 764: WT should be PKO, LKO should PGKO.

Reply: The text is edited in the revised manuscript.

The description of supplemental figure 4g is incorrect. Should be PPARA instead of PDK4.

Reply: The text is edited in the revised manuscript.

The lack of increase in fasting induced ketone body production by loss of Gps2 in the absence of PPAR α is an important result that should be in the main figures. It is the key result that illustrates the physiological relevance of the purported interaction between PPAR α and GPS2.

Reply: We move the ketone body data to Figure 1b and Figure 3h.

Supplemental figure 5i are not publication quality. Please be gentle to the eyes of the reviewer.

Reply: We improved the figure quality in the revised manuscript (Supplementary Figure 4m-n).

Scientific writing does not allow for the use of “proof” and “prove”. Please use more scientific alternatives (evidence, demonstrate).

Reply: We edited the revised manuscript accordingly.

The Co-IP data showing a physical interaction between PPAR α and GPS2 should be moved to the main figures.

Reply: We moved the IP data to Figure 3f as suggested.

Reviewer #3 (Remarks to the Author):

Liang et al. submit a manuscript entitled “GPS2 accelerates the progression of nonalcoholic steatohepatitis through PPAR α -selective mechanisms”. They first identify GPS2, a subunit of the NCoR/HDAC corepressor complex, in an analysis of published RNA-seq data from human liver biopsies. Even though GPS2 expression did not itself correlate significantly with NAFLD/NASH status, they noted that its expression nonetheless correlated with expression of fibrogenic genes. The authors then move to mouse models with liver-specific deletion (LKO) of GPS2 driven by albumin-Cre. When these mice were subject to MCD diet, the LKO mice were protected from liver steatosis, damage, and fibrosis, with gene expression suggesting higher fatty acid oxidation (FAO). When these mice were subjected to HFD, they were protected from body weight gain and, consistent with this, there was improvement in glucose tolerance and insulin sensitivity, as well as liver and serum TGs. RNA-seq of WT vs LKO mice showed dysregulation of multiple lipid metabolic pathways, and the authors focus on PPAR α as a regulator of FAO. Livers from LKO mice showed increased induction of PPAR α target genes upon fasting or treatment with the PPAR α agonist GW7647, yet no change in gene activation by an LXR agonist. Furthermore, PPAR α occupancy by CHIP-qPCR at two strong binding sites (Pdk4 and Cyp4a14) was increased in LKO livers. In contrast to GPS2 LKO mice, double knockout mice lacking liver GPS2 and whole body PPAR α failed to show increased expression of PPAR α target genes, showing that GPS2 deficiency requires PPAR α to induce these genes. CHIP-seq for GPS2 was performed, showing enrichment of DR-1 PPAR motifs and occupancy at sites with enhancer and promoter histone marks, and changes in these marks upon GPS2 knockout that correlate with nearby gene expression. GPS2 recruitment to sites near Pdk4 and Cyp4a13 was reduced in livers lacking NCoR (NCOR1) but not SMRT (NCOR2), indicating an interesting selectivity of GPS2 for the NCOR1 corepressor complex. The overall mechanism proposed is that GPS2 knockout de-represses PPAR α to activate FAO and ketogenesis, thus decreasing liver steatosis and fibrosis.

This manuscript is overall well-written and very interesting. The statistical tests appear appropriate, as does the detail in the methods. The fact that GPS2 LKO protects against hepatic steatosis, while knockout of other subunits of the NCoR/HDAC3 complex cause steatosis, is alone a novel and exciting result. The authors present some very good evidence supporting the mechanism of de-repression of PPAR α target genes, but additional key experiments (see below) would further bolster the model.

General comments:

1) The first part of the paper in human liver samples nicely shows variable expression of GPS2. Are the genetic variants (i.e. eQTLs) associated with GPS expression?

The authors should interrogate the GTEx dataset to address this.

Reply: We explored the GPS2 eQTLs in ExSNPs from 16 publicly available human eQTL human studies. We found rs2270339 a potential GPS2 eQTL in human livers. However, we do not have clinical resources to verify this eQTL, therefore we decide not to include this data in the revised manuscript.

The other possibility is that GPS2 expression changes in the setting of liver steatosis. For this reason, the authors should report whether mouse liver GPS2 expression changes in their MCD and HFD models.

Reply: We have performed the qPCR analysis in HFD and MCD mice and included the data in **Supplementary Figure 1j** and **Supplementary Figure 2j**.

2) The GPS2 LKO mouse model is key to this manuscript, as is the successful ChIP for GPS2 in mouse liver. The tissue-specific knockout of GPS2 is shown nicely at the mRNA level and protein levels (Fig 2B-C), and should also be shown at the level of the ChIP assay (i.e. GPS2 ChIP occupancy is decreased/absent in the LKO). Furthermore, the authors state that other components of the NCOR/HDAC3 complex do not change upon GPS2 LKO based on mRNA levels in Supplementary Fig 2B, but to make this conclusion Western blots are necessary to show unchanged protein levels.

Reply: We have performed now the ChIP-seq of GPS2 in WT and LKO mice and include the tracks in **Figure 5c-d**. We have also performed the HDAC3 western blot in WT and LKO mice and include the data in **Supplement Figure 1b**. We are not able to obtain sufficient quality western blot data for NCOR and other subunits. However, we have performed NCOR ChIP-seq in WT and LKO mice (**Figure 5** and **Supplemental Figure 5**) and see no differences in recruitment, suggesting NCOR levels not to be affected by GPS2 depletion.

3) For many of the bar graphs (i.e. liver TG in Fig 2G and 3H, etc), it would be preferable to show the individual data points for each mouse as well as the mean/sem. I'm not sure whether Nature Communications has this requirement, but it's generally considered good practice today.

Reply: We have revised all the panels accordingly.

4) The serum ketone data (Supp Fig 2G and 5E) is impressive and highly relevant, as PPAR α is the key regulator of ketogenesis. I recommend moving this data to the main rather than supplemental figures. (Also adding *Hmgcs2* to panel of mRNAs measured would bolster this.) The changes in ketones are the only functional readout for the proposed mechanism: that GPS2 knockout de-represses PPAR α to activate FAO and ketogenesis, thus decreasing liver fat and VLDL secretion. Ideally, one other other functional assay supporting this model should be included (i.e. direct measurement of FAO, indirect measurement like acylcarnitines, measurement of VLDL secretion, etc).

Reply: We moved the ketone body data to **Figure 1b** and **Figure 3h**. We include now an *in vitro* Seahorse assay in WT and GPS2 knockdown cells, and observed enhanced FAO by GPS2 knockdown (**Figure 1c**).

5) The authors propose that GPS2 in liver lacks the anti-inflammatory properties it has in other tissues, but this was only shown after the major inflammatory stimulus of LPS treatment. Liver inflammation assays should also be reported in their MCD and HFD models of more “metabolic inflammation.”

Reply: We include now the F4/80 staining of both MCD and HFD mice in Figure 1k, 1l and Figure 2i, 2j. Analysis of inflammatory gene expression (F4/80) is shown in Figure 1m and Figure 2k.

Major issues:

1) In the MCD experiments in Figures 2D-H, a control diet group is missing. For all the improvements seen in GPS2 LKO mice (lower transaminases, altered gene expression, liver TGs, Sirius red), it's unclear how much improvement occurred and whether it approaches normal liver. If this data exists, I recommend inclusion (the gene expression differences on a control diet are indeed in supp Fig 2M).

Reply: We now add the control group in the revised manuscript in Figure 1b-m, Figure 2b-k.

Another issue is the panel of genes in Figure 2F.

Were other fibrotic and lipid metabolic genes tested and showed no difference? These could be shown in supplemental.

Reply: We tested the same gene panels in MCD and HFD mice, and include the data in Supplementary Figure 2p.

Furthermore, this panel lacks any inflammatory genes, and inflammation is a key aspect of NASH that is induced in the MCD model. Therefore, it would be of great value to include inflammatory genes like ILs, TNF, etc in this panel (see comment 5 above). A key point in Supp Fig 2I is that these genes did not differ upon LPS treatment (so GPS2 has a selective role in fibrosis), but it is critical to know in the MCD model whether GPS2 LKO is anti-inflammatory.

Reply: We include the panel of pro-inflammatory genes in Figure 1m and Figure 2k.

Finally, for the key point of reduced fibrosis, some quantification is necessary beyond the gene expression in Fig 2F and the staining in Fig 2H. The Sirius red staining (% of area) is routinely quantified, and hydroxyproline is a standard biochemical assay of collagen deposition.

Reply: We include the Sirius red staining and hydroxyproline data in Figure 1h-j.

2) In the text describing the HFD diet model in Figure 3, it must be more explicitly acknowledged that the decreased body weight and adiposity of the LKO mice is alone consistent with all the metabolic improvements observed. Therefore, the concluding sentence (line 214-215) that hepatocyte GPS2 modulates lipid/glucose homeostasis needs to be qualified by stating that weight is reduced. It is an interesting question but beyond the scope of this study to determine why less weight is gained (i.e. metabolic cage analysis of energy expenditure, etc).

Reply: The conclusion of Figure 3 is revised according to the suggestion, adding hepatocyte GPS2 ‘knockout reduces body weight’ in the sentence.

We hypothesize that the reduced adipose tissue mass and body weight gain is largely due to enhanced *Fgf21* expression in *Gps2* LKO mice. As FGF21 promotes adipose tissue browning

and promotes fatty acid catabolism. Indeed, basal FGF21 expression in the LKO mice almost reaches the fasting or GW (PPAR α agonist) levels (**Figure 3c and 3d**). FGF21 is a direct target of GPS2 which is dependent on PPAR α (**Figure 3g, Supplementary Figure 3b-c**). Indeed, LKO mice in the HFD mice showed increased expression of adipose tissue (VAT and SAT) browning markers (see the attached).

Finally, given that liver fibrosis is described as the key role of GPS2, no assays of fibrosis (gene expression, staining) were reported on this HFD model. This is likely because not much fibrosis is induced by this diet, but the point should at least be mentioned (“In contrast to the MCD, no difference in fibrotic signature was observed in GPS2 LKO in this HFD model, which failed to trigger much fibrosis.”)

Reply: The fibrosis in the HFD-treated mice is marginal, we tested the fibrosis genes and include the data in **Supplemental Figure 2p** of the revised manuscript. The main text is revised accordingly.

3) The PPAR α ChIP data in Fig 4H-I is missing the key negative control. While IgG is okay, it would be preferable to show the PPAR α ChIP signal is at a background genomic site without any binding. Especially given the low percentage of input at such strong sites (Pdk4 and Cyp4a14), Q-PCR primers for a negative control site are essential here. Another potential negative control for the PPAR α ChIP is using the PPAR α knockout livers described later. Likewise, other ChIP-QPCR experiments in Figure 5-6 would also benefit from the negative control of a non-binding site, to see the degree of enrichment of binding sites over background in the ChIP DNA.

Reply: We have successfully optimized the PPAR α ChIP protocol (which included the identification of a new commercial antibody as alternative to the discontinued Santa Cruz antibody) and performed a new PPAR α ChIP-seq analysis in GPS2, NCOR and PPAR α KO livers, along with new ChIP-seq data for GPS2 and NCOR (**Figure 5c-d**). These new data provide the missing controls (i.e. PPAR α ChIP-seq in PPAR α KO mice) and strongly support the specificity of the ChIP/ChIP-seq data. Therefore, we have removed the old ChIP-qPCR data (old Figure 4H-I). We have now included the negative control region in the revised ChIP-qPCR data in **Figure 4G** and **Supplementary Figure 5i**.

4) Figure 5 is confusing and mostly unnecessary. The figure title says that “GPS2 deletion causes epigenetic activation”, but the data show apparently similar amounts of “inactivation” (i.e. genes and histone marks that go down with GPS2 deletion). The correlations in Figures 5C-H simply show that changes (up or down) in activating histone marks and nearby gene expression generally correlate with one another (globally and at individual loci), which is always seen and reveals nothing about GPS2 biology. Such data could be supplemental. The most potentially relevant result with PPAR α knockout is in Fig 5I, but has the same issue as above missing the negative control site without H3K27ac, and even more problematic it’s

missing the key groups with PPAR α present (and presumably higher H3K27ac at these strong target genes). To correctly interpret this experiment two more groups are needed: wild type and GPS2 deletion only (comparing these to show that there's a statistically significant increase in acetylation detectable by this ChIP QPCR assay). Without this, it's difficult to make the conclusion that GPS2 needs PPAR α to affect the epigenome. Rather than just looking at the epigenomic markers, an even better test of the authors' model would be looking at GPS2 genomic occupancy in their same PPAR α knockout mice – ideally by ChIP-seq but at least by ChIP-QPCR at selected sites (analogous to the GPS2 ChIP in the NCOR1/2 knockout mice in Fig 6E). **Showing a loss of GPS2 occupancy in the absence of PPAR α is essential to the overall model.**

Reply: We acknowledge the comments for improvements and clarifications and have addressed them in the revision.

Most importantly, we generated new ChIP-seq data for PPAR α , GPS2 and NCOR in WT and PPAR KO livers (as well as in GPS2 and NCOR ko livers, extensively discussed in our response to Reviewer 1). These data indeed demonstrate loss of GPS2 occupancy, along with loss of NCOR, in the absence of PPAR α (**Figure 5, Supplementary Figure 5**). Interestingly, GPS2 and especially NCOR seem not completely abolished from all PPAR-target loci (e.g. at *Pdk4* and *Cyp14a14*). This would be consistent with the possibility that other liver TFs/nuclear receptors occupy and recruit NCOR/GPS2 (likely the HDAC3 co-repressor complex) to these sites in the PPAR α KO mice (possibly HNF4, *Armour SM et al Nat Commun 8, 2017*).

Re the role of GPS2 in the epigenomic activation of PPAR α target loci and genes: We believe that the increase of H3K27ac signals at PPAR α target loci, in conjunction with the revised data in Figure 5 (extensively discussed in our response to Reviewer 1), provide supportive evidence for the model that loss of GPS2-PPAR α interactions (involving also NCOR) leads to de-repression of transcription at PPAR α -target gene loci. This model is limited to repressed PPAR α -target genes, while other mechanisms may account for the GPS2-dependent but PPAR α -independent genes (e.g. those that are down-regulated upon GPS2 depletion but not linked to PPAR α).

We have revised the titles of Figure 4 and Supplementary Figure 4 to better express this.

We have additionally performed new H3K27ac ChIP-seq in WT, *Gps2* LKO, *Ncor* LKO and *Smrt* LKO livers. We observed a similar H3K27ac increase at *Pdk4* and *Cyp4a14* loci in *Gps2* and *Ncor* LKO livers but not *Smrt* LKO livers (**Figure 5a-b**). In conjunction with the new ChIP-seq data in Figure 5c-d we think that there is a meaningful link between loss-of-GPS2/NCOR and increase of H3K27ac.

In line with the reviewers suggestion we have now included the WT and *Gps2* LKO controls in the H3K27ac ChIP-qPCR analysis (old Figure 5i, now revised **Figure 4g**). Again, these data support the connection of PPAR α and GPS2 in specifying the level of H3K27ac at PPAR α target loci.

The analysis of the H3K27ac/H3K4me3 ChIP-seq data is now moved to **Supplementary Figure 4i-4l** of the revised manuscript.

Minor issues:

line 2: In the title, I recommend “PPAR α -dependent” rather than “-selective”. A lot of data in double knockouts supports the necessity of PPAR α , but only the supplemental experiment with an LXR agonist (Suppl Fig 4H) supports selectivity, and other nuclear receptors and transcription factors are not addressed.

Reply: The title is revised to ‘Hepatocyte-specific loss of GPS2 in mice reduces non-alcoholic steatohepatitis via activation of PPAR α ’.

line 28: in the abstract, the authors should clearly identify GPS as “a subunit of the NCOR/HDAC3 nuclear receptor corepressor complex”, or something to that effect.

Reply: We revised the abstract as suggested.

line 74: rather than “a fundamental co-repressor complex”, the name “NCOR/HDAC3 corepressor complex” should be used instead (like it is in the discussion), with mention that it’s the major corepressor for nuclear receptors, and also for other TFs.

Reply: We revised the manuscript as suggested.

lines 76-80: I would identify NCOR1/2 as the main structural components of the complex, HDAC3 as the catalytic one, and TBLs and GPS2 as other small subunits of uncertain function – then this manuscript does a nice job ascribing some function to GPS2.

Reply: The text is revised accordingly in introduction and discussion.

line 122: This sentence identifies a human NASH signature gene set. Was this gene list published in the original papers? Or did it emerge from this re-analysis? If the latter, then it would be useful to include the gene lists in a supplementary table.

Reply: This gene list is published on the previous paper, we highlighted it in the revised manuscript.

line 123: The critical result that “expression of GPS2 was not different in NASH liver” is stated but no data is shown. The (negative) data supporting this important point should at least be in the supplemental material.

Reply: We included the data in the **Supplementary Figure 6a.**

line 124: Even though the list of GPS-correlated genes is on the x-axis of Fig 1B, it would also be useful to put these in a supplemental table.

Reply: The list of the genes is from previously published study, we highlighted the source of the gene list in the revised manuscript. The genes correlated with GPS2 are now listed in the **Supplementary Table 2.**

line 139: In Figure 1G, it is unclear whether the x-axis refers to GPS2 mRNA levels before or after weight loss. The same issue applies to the analyses in Suppl Figures 1F-I.

Reply: We deleted old Supplementary Figure 1f-1i due to weak correlations (as suggested by the other reviewers) in the revised manuscript.

line 149: The call-out for tissue-specific knockout say Supp Fig 2B, but it should refer to main Fig 2B.

Reply: The text is edited in the revised manuscript.

line 150: There is no description of data in Supp Fig 2B, so the end of this sentence should

include a clause like “without effect on mRNA levels for other subunits of the NCoR/HDAC3 complex (Supp Fig 2B).”

Reply: The text is edited in the revised manuscript.

line 152: Rather than “lipoprotein profiles”, just state that TG in the VLDL fraction and the total TG were decreased.

Reply: The text is revised accordingly.

line 158: conclude the sentence about unchanged lipases with the explanation: “suggesting decreased hepatic VLDL production rather than increased lipase activity.”

Reply: The text is revised accordingly.

line 220: For this RNA-seq, I presume the mice were on control/chow diets but it’s not apparent from the text, figure legend, or methods. Also, in describing other transcriptome data the authors nicely give the number of genes up- and down-regulated. It looks like a lot based on the heatmap in Fig 4A, but numbers and a gene list in the supplementary data are advised.

Reply: The description of the RNA-seq is added to the text. Due to the large number of up/down regulated genes, we attached the analysed RNA-seq data in the GEO database.

line 239: the text says “enhanced response to...agonist” yet the data in Figure 4D is only the presence of agonist. It would be better for this graph to include the untreated group to see the induction by agonist in each genotype. Also, the y-axis is “relative gene expression” which indicates that one condition has been set equal to 1 for each gene (i.e. in Fig 4D it appears that WT fed is the normalizer). Especially from the Pdk4 data it’s clear that there’s no normalizer condition on the current graph in 4D.

Reply: The old Figure 4D is normalized to untreated WT control, we previously did not include the untreated group. In the revised manuscript, we include all the four groups as suggested by the reviewer (**Figure 3c-d**).

line 244: The PPARa CHIP-seq data in Fig 4F-G appear at first as it was done by the authors, but the methods make clear it was published by others, therefore the citation to Lee et al, Nature 516:112 should be included. Furthermore, browser tracks at two loci are used to make the case that GPS2 and PPARa co-occupy sites in liver. A global analysis of the CHIP-seq peaks (i.e. Venn diagram overlap, scatterplot, etc) would be better to make this point (i.e. “85% of PPARa peaks identified by others show GPS binding signal in our data”).

Reply: We have cited now the reference in the revised Results, in **Figure 4d-e** legend and in the Methods (GO number). For the Venn diagram in **Figure 4f** of the revised manuscript we have now used our own newly generated CHIP-seq data for PPAR and GPS2.

Line 255: for clarity in this section, “LKO” should be qualified as “GPS2 LKO” to avoid confusion with the PPARa whole body knockout.

Reply: The text is revised accordingly.

line 256: wrong punctuation in middle of line, comma not period.

Reply: The text is revised accordingly.

line 268: PDK4 is the only gene shown to make the case that GPS2 express negatively correlates with PPARa target gene expression in human liver. There are many other classic PPARa target genes measured on these arrays that could be correlated to make this this case. It would be preferable to see a heatmap-type analysis showing correlation of multiple PPARa

targets with GPS2 expression.

Reply: In revised **Figure 6h**, we performed a correlation analysis of GPS2 and PPARA separately with NASH correlated genes, and the data indicate largely negative correlation profiles, suggesting that GPS2 and PPARA regulate genes in an opposite manner.

line 417: The word “in” should not be italicized.

Reply: Revised as suggested.

In Fig 2G, is the y-axis correct? 800ug/mg implies that the liver is 80% fat by weight?

Another reason to include control diet livers in this analysis as above.

Reply: 800 ug/mg refers to 800 ug of lipid per mg of liver protein (measured by BCA assay). We include the control diet liver TG in the revised manuscript (**Figure 1g and 2h**) and re-label the y axis to ‘Liver TG ug/mg protein).

In supplementary figure 5, there are two “E” panels and no “D”.

Reply: The revised manuscript is adjusted accordingly.

In supplementary figure 4, panels G and H should be later to preserve the order they are described in the text.

Reply: The revised manuscript is revised accordingly.

Reviewers' comments:

Reviewer #1 (Remarks to the Author):

The revised manuscript has been improved with the newly obtained results. Most of my previous concerns have been resolved. However, there is an issue with the new data (Fig. 5e), which provided evidence for the formation of a stable ternary complex between GPS2, PPAR α , and NCoR. In this figure, the authors showed that NCoR can pull down more PPAR α in the presence of GPS2. Although the results suggest that the three proteins interact with each other to form a stable complex, another explanation is that the increased PPAR α signal could reflect an additive effect of two different complexes: one involves direct NCoR/PPAR α interaction and the other involves indirect NCoR/PPAR α interaction through GPS2. Can the authors further clarify these different possibilities, given that it is one of the key experiments for the revised manuscript? The authors should also discuss in more details how GPS2, NCoR, and PPAR α interact to form a stable ternary complex. For example, does this involve simultaneous interactions of PPAR α with NCoR and GPS2?

Another issue of Fig. 5e is that the authors need to show immunoprecipitated NCoR signals to rule out the possibility that the increased PPAR α signal was due to a higher pull-down efficiency.

Reviewer #2 (Remarks to the Author):

The manuscript has been vastly improved. Please consider the following remaining comments.

Please remove figure 6g. It does not add anything to the paper.

The sentence on line 260 ""Both GPS2 and PPAR α recruitment in the Pdk4 and Cyp4a14 loci (Fig. 4d-e, GSE61817)" lacks a verb. Please adjust.

Signed, Sander Kersten

Reviewer #3 (Remarks to the Author):

Liang et al. submit a revised manuscript now entitled “Hepatocyte-specific loss of GPS2 in mice reduces non-alcoholic steatohepatitis via activation of PPAR α ”. The revised manuscript does an excellent job addressing my critiques of the initial submission. In particular, the new ChIP-seq data is impressive and bolsters their model for mechanism. De-emphasizing and moving the human data to the end is another major improvement. This manuscript is suitable to publication once the authors address one major and a few minor issues:

Figures 4-5: To the authors’ great credit, the new manuscript includes several new ChIP-seq datasets nicely generated in the relevant knockout models, but analysis of these is generally limited to individual loci rather than genome-wide. For instance, there is quantification of ChIP-seq peak data: i.e. in Fig 4b, H3K27ac goes up 33% at the Pdk4 TSS in LKO mice. The legend describes a fold change based on biological duplicates, so statistics cannot be performed, and it seems these numbers are simply there to quantify the difference in peaks on the browser and apparent by eye. This is fine if that is acknowledged, and it would be much better to show that these sometimes-small differences are indeed reproducible and statistically significant, in the ChIP-seq or in qPCR experiments. Another way to show significance would be probing the global genome-wide effects, versus only those on the Pdk4 and Cyp4a14 loci (and Acot in supplemental) that are shown. Are these changes in histone marks and factor occupancy at these loci representative of genome-wide effects, or are they selective? For instance, in Figure 5c-d (and supplemental 5g), it looks quite impressive that GPS2 ChIP signal is lost in Ncor1 LKO, yet not vice versa, and this result is highly relevant to the authors’ model. Is this only at these 3 sites, or everywhere in the genome? If the latter, then how can the authors rule out a low efficiency ChIP reaction (due to technical variability in the method) versus a global decline in occupancy? The authors did a lot of hard work to generate this new ChIP-seq data, and presenting it only as browser tracks at selective loci is not adequate without some efforts at genome-wide integration.

Minor issues:

Line 52: change “in consistence” to “consistent”

Line 65: delete words “to be”

Line 155: call out should refer to supplementary not main figure

Line 166: change “months” to “month”

Line 167-174: Please revise to make diet clearer. The description of data in supplementary figures 2a-2i is confusing. It appears that all this data was in mice fed a standard chow diet, correct? Yet the introductory sentence for this section (line 165-166) is about HFD, as is the rest of the paragraph (lines 175-192). This section needs to more clearly progress from data on chow diet, leading to further investigation on HFD.

Line 200: change “feeding” to “fed”

Line 261-262: rephrase this sentence. Based on the Venn diagram in Fig 4f, only ~66% of PPAR α sites overlap GPS2. However, taking the smaller GPS2 cistrome as the denominator allows the even stronger conclusion that “over 85% of GPS2 sites identified here co-localized with previously reported PPAR α sites”.

Line 300-301: This new Pol2 ChIP-seq data is out of place here, as a separate figure 5f. Pol2 occupancy correlates with gene expression and histone mark data, and does not specifically belong in this context regarding protein-protein interactions in the complex. Please move these browser tracks to the bottom of Fig 5c-d, where they can be described as consistent changes in Pol2 occupancy related to all the other changes.

Line 381: delete word “the”

Line 383-384: I think this is backwards. Lack of CETP gives mice higher HDL-C and lower VLDL than humans.

Line 693: The figure 2 legend should say “...HFD-induced weight gain, insulin resistance, and...”

Lines 718-722: In panel 3A, the inset of the heat map is unclear. Are these a set of PPAR α targets? If so, are they just selected/illustrative examples, or a list derived from the analysis in supplementary figure 3?

Figure S1F: change “HPL” to “HL” as abbreviation for hepatic lipase.

Itemized list of all changes made in the revised manuscript

Revised Fig.5c-d (old Fig.5c-d): We performed additional PPAR α ChIP-seq in WT and GPS2 LKO livers and performed a statistical significant analysis of the highlighted peaks (labelled with *) from all ChIP-seqs in biological triplicates (n=3 mice). We also combined the PolII ChIP-seq (old Fig.5f) to Fig.5c-d.

Added Fig.5e-g: We have performed genome-wide analysis of GPS2 ChIP-seq in NCOR and PPAR α LKO livers, which showed globally reduced GPS2 recruitment in both peak numbers and peak intensity.

Added supplementary Fig.5j-l: We have performed global analysis of PPAR α ChIP-seq in NCOR and GPS2 LKO livers. Average PPAR α recruitment is increased in both LKO livers, the enhanced peak numbers also exceed the reduced ones, especially in GPS2 KO livers, consistent with our model.

Revised supplementary Fig.6d (old Fig.5e): We add the NCOR western blot in the immunoprecipitated samples (from the same batch of experiment) to exclude the different IP efficiency during the experiment.

Deleted old Fig.6g: The figure is deleted according to the reviewer 2 suggestion.

Point-by-point response to the reviewer's comments

Our responses are indicated in RED.

Reviewers' comments:

Reviewer #1 (Remarks to the Author):

The revised manuscript has been improved with the newly obtained results. Most of my previous concerns have been resolved. However, there is an issue with the new data (Fig. 5e), which provided evidence for the formation of a stable ternary complex between GPS2, PPAR α , and NCoR. In this figure, the authors showed that NCoR can pull down more PPAR α in the presence of GPS2. Although the results suggest that the three proteins interact with each other to form a stable complex, another explanation is that the increased PPAR α signal could reflect an additive effect of two different complexes: one involves direct NCoR/PPAR α interaction and the other involves indirect NCoR/PPAR α interaction through GPS2. Can the authors further clarify these different possibilities, given that it is one of the key experiments for the revised manuscript? The authors should also discuss in more details how GPS2, NCoR, and PPAR α interact to form a stable ternary complex. For example, does this involve simultaneous interactions of PPAR α with NCoR and GPS2?

Reply: We appreciate that the reviewer raised this intriguing mechanistic issue. We have now further revised our discussion to better clarify our model how we believe GPS2 and NCOR act together to repress PPAR α activity. In our discussion we refer to detailed mechanistic *in vitro* studies (e.g. the PPAR γ study Guo *et al. J Biol Chem* 290, 3666-3679, 2015, our previous LXR study Jakobsson *et al., Mol Cell* 34, 510-518, 2009), the results of which may help to better understand the PPAR α interactions described here.

There is published evidence that GPS2 forms a stable core complex with NCOR/SMRT and TBL/TBLR1 as core subunit of the corepressor complex, supported by NMR structure data (Oberoi *et al. Nat Struct Mol Biol* 18, 177-184, 2011) and IP-mass spec data (e.g. Zhang *et al., Mol Cell* 9, 611-623, 2002; Armour *et al., Nature Communications* 8, 549, 2017). **However, there is so far no direct evidence whether this 'corepressor complex core' forms stable ternary complexes with nuclear receptors, including PPAR α .** The current data, including those presented in our study, rather suggest that repression *in vivo* results from dynamic (perhaps cycling), weaker, transient interactions of the corepressor complex with transcription factor targets such as PPAR α . NCOR and GPS2 are proposed to be largely intrinsically disordered in nature, which may reflect the involvement in many specific but relatively low affinity interactions. These aspects are extensively discussed in reviews from the John Schwabe (Watson *et al., Mol Cell Endocrinol.* 348: 440-449, 2012) and Mitch Lazar laboratories (Emmett & Lazar *Nat Rev Mol Cell Biol*, 2018 *in press*).

NCOR has been previously known to be the main receptor (also PPAR α) binding subunit of the complex, and our new data do not challenge this view. Indeed, our ChIP-seq data confirm that most GPS2 interactions with PPAR α at chromatin are lost in NCOR KO hepatocytes, while NCOR interactions were not abolished upon in *Gps2* KO hepatocytes. **These results are critical, as they indicate that NCOR does not require GPS2 to interact with PPAR α *in vivo*,** thus contrasting the GPS2-NCOR bridging model suggested for agonist-bound PPAR γ (Guo *et al. J Biol Chem* 290, 3666-3679, 2015).

Our new work identifies with GPS2 an additional core subunit of the complex that also interacts with PPAR α and contributes to the overall repression *in vivo*. The Co-IP data and the inclusion of GPS2 mutants (that are deficient in interactions with either NCOR or PPAR α) add *in vitro* support that GPS2 utilizes different domains to bind PPAR α vs. NCOR and that this binding may be cooperative to increase PPAR α interactions with NCOR. This is fully compatible with the above outlined transient interaction model where cooperative interactions of two subunits explain why depletion of either NCOR or GPS2 reduces PPAR α repression capacity and target gene expression *in vivo*.

Regarding the details of the PPAR α interactions with GPS2 and NCOR, we believe that any of the possible different interaction modes are compatible with the model and the *in vivo* data we present in this study. Based on the published *in vitro* data, NCOR and GPS2 interact with distinct surfaces at the ligand-binding domain (LBD) of nuclear receptors. In case of NCOR, the receptor interaction domain (RID) is largely unstructured but adopts upon interaction a helical peptide structure (the CORNR box) that interacts with the conserved cofactor surface at the LBD helices 3-5 (but excludes AF-2 helix 12), see for example the structure of the PPAR α LBD with a corepressor peptide in the presence of an PPAR antagonist (Xu *et al.*, *Nature* 415, 813-817, 2002; Dowell *et al.*, *J Biol Chem* 274, 15901-7, 1999; Liu *et al.*, *Mol Endocrinol* 22, 1078-92, 2008). Although there is no structure data for any receptor (including PPAR α) interaction domain of GPS2 (aa 105-155, this study), *in vitro* assays using receptor mutations and NCOR peptide competition strongly suggest that GPS2 binds to a surface that is different from the AF-2/NCOR-binding surface. In particular, the GPS2 surface seems exposed irrespective of the ligand status (agonists vs. antagonists), marking a fundamental difference to the classic coactivator (LXXLL motif) corepressor (CORNR motif) surfaces. This has for example in detailed been studied *in vitro* for PPAR γ (Guo *et al.* *J Biol Chem* 290, 3666-3679, 2015) and by us for LXR α (Jakobsson *et al.*, *Mol Cell* 34, 510-518, 2009), and thus appears likely to be relevant for PPAR α /RXR. One interesting *in vivo* scenario is that activation by endogenous PPAR α agonists may lead to selective disruption of NCOR interactions, while GPS2 interactions may serve to stabilize the corepressor complex interactions, explaining the effects of the GPS2 LKO alone. We include this into the discussion of the current manuscript.

In our opinion a meaningful further experimental analysis of the distinct possible interaction modes (ternary complexes, independent binary interactions, heterodimer interactions) **cannot be achieved within this study** and deserves a focussed future investigation. In particular, to be relevant for the *in vivo* situation and the ChIP-seq data on chromatin, one would need to study the issue in the context of the PPAR α /RXR heterodimer dimer on DNA, including full-length GPS2 and NCOR proteins. This is currently impossible *in vitro*: for example, in our previous GPS2-LXR study (Jakobsson *et al.*, *Mol Cell* 34, 510-518, 2009) we had access to the purified LXR/RXR heterodimer (a co-author purified the proteins for structure analysis over several years), which we could reconstitute on DNA to map the interactions with purified GPS2 protein. We also envisage *in vivo* studies where we would reconstitute GPS2 mutants (such as those described in our study, NCOR dead vs. PPAR α dead mutants, preferentially point mutations not changing other GPS2 features) in hepatocytes of GPS2 LKO mice using virus (AAV)-delivery, but such experiments require far more time, tools to be generated, and permissions to be obtained, thus not possible within a revision.

Another issue of Fig. 5e is that the authors need to show immunoprecipitated NCoR signals to rule out the possibility that the increased PPAR α signal was due to a higher pull-down efficiency.

Reply: We have revised Old Fig. 5e (**now revised supplementary Fig. 6d**) as suggested by including the western blot of immunoprecipitated NCOR from the same IP samples (lower right panel, using anti-FLAG antibody to detect FLAG-NCOR).

Reviewer #2 (Remarks to the Author):

The manuscript has been vastly improved. Please consider the following remaining comments.

Please remove figure 6g. It does not add anything to the paper.

Reply: Fig.6g is removed in the current manuscript.

The sentence on line 260 “Both GPS2 and PPAR α recruitment in the Pdk4 and Cyp4a14 loci (Fig. 4d-e, GSE61817)” lacks a verb. Please adjust.

Reply: The sentence is changed accordingly.

Signed, Sander Kersten

Reviewer #3 (Remarks to the Author):

Liang et al. submit a revised manuscript now entitled “Hepatocyte-specific loss of GPS2 in mice reduces non-alcoholic steatohepatitis via activation of PPAR α ”. The revised manuscript does an excellent job addressing my critiques of the initial submission. In particular, the new ChIP-seq data is impressive and bolsters their model for mechanism. De-emphasizing and moving the human data to the end is another major improvement. This manuscript is suitable to publication once the authors address one major and a few minor issues:

Figures 4-5: To the authors’ great credit, the new manuscript includes several new ChIP-seq datasets nicely generated in the relevant knockout models, but analysis of these is generally limited to individual loci rather than genome-wide. For instance, there is quantification of ChIP-seq peak data: i.e. in Fig 4b, H3K27ac goes up 33% at the Pdk4 TSS in LKO mice. The legend describes a fold change based on biological duplicates, so statistics cannot be performed, and it seems these numbers are simply there to quantify the difference in peaks on the browser and apparent by eye. This is fine if that is acknowledged, and it would be much better to show that these sometimes-small differences are indeed reproducible and statistically significant, in the ChIP-seq or in qPCR experiments. Another way to show significance would be probing the global genome-wides effects, versus only those on the Pdk4 and Cyp4a14 loci (and Acot in supplemental) that are shown. Are these changes in histone marks and factor occupancy at these loci representative of genome-wide effects, or are they selective?

Reply: We thank the reviewer for the comments to have now further improved our study by including a genome-wide analysis of the ChIP-seq data and confirmation via independent ChIP-qPCR analysis.

The histone modification (H3K27ac and H3K4me3) ChIP-seq data are based on duplicate experiments (2 mice each), therefore H3K27ac changes were further validated in independent ChIP-qPCR experiments (Fig. 4g) (n = 3-5 mice in each group).

We additionally performed global analysis of these histone modification changes at the loci of GPS2 target genes (Supplementary Fig.4i-4l). The analysis revealed that H3K27ac and H3K4me3 levels were significantly changed at GPS2-repressed or activated gene loci (defined by upregulated or downregulated genes in GPS2 LKO livers, as compared to WT) comparing with non-target loci (unchanged genes). This suggests that the histone modification changes upon GPS2 LKO are not only limited to the *Pdk4* and *Cyp4a14* loci but reflects global epigenome changes consistent with the transcriptome changes upon GPS2 removal.

For instance, in Figure 5c-d (and supplemental 5g), it looks quite impressive that GPS2 ChIP signal is lost in *Ncor1* LKO, yet not vice versa, and this result is highly relevant to the authors' model. Is this only at these 3 sites, or everywhere in the genome?

Reply: We now performed the genome-wide analysis of GPS2 ChIP-seq in NCOR LKO and PPAR α KO livers and found that the reduced recruitment of GPS2 upon NCOR or PPAR α removal is not limited to these selected regions. We now inserted these new analysis into the current manuscript (Fig.5e-5g, Supplementary Fig.5j-5l).

First, average tag counts (logFC) of GPS2/NCOR (or GPS2/PPAR α) co-localized peaks were significantly reduced in the respective KO livers (Fig.5e). There are also far more reduced than increased GPS2 peaks in both NCOR LKO and PPAR α KO livers comparing with WT mice (Fig.5f and 5g). **These data indicate a genome-wide effect of either NCOR or PPAR α depletion on GPS2 chromatin recruitment in hepatocytes.**

Second, we had in the meantime generated additional PPAR α ChIP-seq data and now include in the current manuscript the analysis of biological triplicates (n=3 mice) in WT and GPS2 LKO livers (the peak tag counts change is revised accordingly). All the changes as labelled in the Fig. 5c and d are now calculated based on the triplicates, and we performed the statistical analysis in those gene loci and inserted the results (using * to represent the statistically significant changes in Fig. 5c and d). The new analysis revealed that the global PPAR α recruitment in NCOR and GPS2 LKO mice was increased (Supplementary Fig. 5j), supporting that GPS2 and NCOR work together (in the complex) to affect PPAR α . There were more increased PPAR α peaks comparing with reduced peaks at the whole genome level in both GPS2 and NCOR LKO livers (Supplementary Fig.5k-5l), further supporting that the changes are not limited to the *Pdk4* and *Cyp4a14* loci.

If the latter, then how can the authors rule out a low efficiency ChIP reaction (due to technical variability in the method) versus a global decline in occupancy? The authors did a lot of hard work to generate this new ChIP-seq data, and presenting it only as browser tracks at selective loci is not adequate without some efforts at genome-wide integration.

Reply: We believe that the changes we report in the ChIP-seq analysis are unlikely due to technical variability for the following reasons:

First, the GPS2 ChIP-seq in NCOR and PPAR α KO livers have been performed as biological TRIPLICATES. In Fig. 5c, 5d and Supplementary Fig. 5g we show representative browser shots for these triplicates. To convince the reviewer, **we show ABOVE all triplicates**. We have deposited the RAW data for all triplicates in our GEO database set.

Second, we have re-analysed the peak intensity changes using triplicates in Fig. 5 c and d. RAW tag counts were normalized first for the calculation the fold change and p value (i.e. we used the whole library size to normalize the technical variation between samples), which is a standard way to minimize the impact of variations during library preparation and sequencing. This analysis using the biological triplicates confirmed the reduction of GPS2 recruitment in both NCOR and PPAR α KO mice. We also compared the changes between each group statistically and include the statistical comparison results in the current manuscript.

Third, the reduced GPS2 recruitment in NCOR LKO livers was confirmed by an independent ChIP-qPCR at *Pdk4* and *Cyp4a14* loci (n=3, Supplementary Fig. 5i), confirming the ChIP-seq results.

Minor issues:

Line 52: change “in consistence” to “consistent”

Reply: ‘In consistence’ is changed to ‘consistent’ in the current manuscript.

Line 65: delete words “to be”

Reply: The words ‘to be’ is deleted.

Line 155: call out should refer to supplementary not main figure

Reply: It refers to the main figure but not supplementary figure.

Line 166: change “months” to “month”

Reply: ‘months’ is changed to ‘month’ in the current manuscript.

Line 167-174: Please revise to make diet clearer. The description of data in supplementary figures 2a-2i is confusing. It appears that all this data was in mice fed a standard chow diet, correct? Yet the introductory sentence for this section (line 165-166) is about HFD, as it the rest of the paragraph (lines 175-192). This section needs to more clearly progress from data on chow diet, leading to further investigation on HFD.

Reply: The text has been edited to more clearly describe the dietary conditions in the current manuscript.

Line 200: change “feeding” to “fed”

Reply: ‘feeding’ is changed to ‘fed’ in the current manuscript.

Line 261-262: rephrase this sentence. Based on the Venn diagram in Fig 4f, only ~66% of PPAR α sites overlap GPS2. However, taking the smaller GPS2 cistrome as the denominator allows the even stronger conclusion that “over 85% of GPS2 sites identified here co-localized with previously reported PPAR α sites”.

Reply: The text is changed according to the reviewer’s suggestions in the current manuscript.

Line 300-301: This new Pol2 ChIP-seq data is out of place here, as a separate figure 5f. Pol2

occupancy correlates with gene expression and histone mark data, and does not specifically belong in this context regarding protein-protein interactions in the complex. Please move these browser tracks to the bottom of Fig 5c-d, where they can be described as consistent changes in Pol2 occupancy related to all the other changes.

Reply: We now move the Pol2 ChIP-seq to Fig. 5c-d in the current manuscript, the text is adjusted accordingly.

Line 381: delete word “the”

Reply: ‘the’ is deleted in the current manuscript.

Line 383-384: I think this is backwards. Lack of CETP gives mice higher HDL-C and lower VLDL than humans.

Reply: We apologize for the mistake, the text is adjusted in the current manuscript.

Line 693: The figure 2 legend should say “...HFD-induced weight gain, insulin resistance, and...”

Reply: Figure 2 legend is adjusted according to the reviewer’s suggestions.

Lines 718-722: In panel 3A, the inset of the heat map is unclear. Are these a set of PPARα targets? If so, are they just selected/illustrative examples, or a list derived from the analysis in supplementary figure 3?

Reply: The heatmap represents the top 1000 significantly changed (sorted by adjusted p-value, both upregulated and downregulated) genes in GPS2 LKO versus WT mice. We now include this description in the Fig. 3a legend.

Figure S1F: change “HPL” to “HL” as abbreviation for hepatic lipase.

Reply: ‘HPL’ is changed to ‘HL’ in the current manuscript.

REVIEWERS' COMMENTS:

Reviewer #1 (Remarks to the Author):

The revised manuscript has improved with newly added results and discussions. There are no further concerns from this reviewer.

Reviewer #3 (Remarks to the Author):

The authors have addressed all my issues. In particular, they did a nice job incorporating replicates, statistics, and genome-wide analyses of their CHIP-seq data. The manuscript is ready for publication in my mind.